# Bulk-surface coupling identifies the mechanistic connection between Min-protein patterns in vivo and in vitro

Fridtjof Brauns [1,4], Grzegorz Pawlik[2,4], Jacob Halatek[3,4], Jacob Kerssemakers[2], Erwin Frey [1✉] & Cees Dekker [2✉]

Self-organisation of Min proteins is responsible for the spatial control of cell division in *Escherichia coli*, and has been studied both in vivo and in vitro. Intriguingly, the protein patterns observed in these settings differ qualitatively and quantitatively. This puzzling dichotomy has not been resolved to date. Using reconstituted proteins in laterally wide microchambers with a well-controlled height, we experimentally show that the Min protein dynamics on the membrane crucially depend on the micro chamber height due to bulk concentration gradients orthogonal to the membrane. A theoretical analysis shows that in vitro patterns at low microchamber height are driven by the same *lateral* oscillation mode as pole-to-pole oscillations in vivo. At larger microchamber height, additional *vertical* oscillation modes set in, marking the transition to a qualitatively different in vitro regime. Our work reveals the qualitatively different mechanisms of mass transport that govern Min protein-patterns for different bulk heights and thus shows that Min patterns in cells are governed by a different mechanism than those in vitro.

[1] Arnold Sommerfeld Center for Theoretical Physics and Center for NanoScience, Department of Physics, Ludwig-Maximilians-Universität München, München, Germany. [2] Department of Bionanoscience, Kavli Institute of Nanoscience Delft, Delft University of Technology, Delft, the Netherlands. [3] Microsoft Research, Cambridge, United Kingdom. [4] These authors contributed equally: Fridtjof Brauns, Grzegorz Pawlik, Jacob Halatek. ✉email: frey@lmu.de; c. dekker@tudelft.nl

The Min-protein system was discovered in *E. coli*[1,2], where pole-to-pole oscillations—that is, the periodically alternating accumulation of the Min proteins on either pole of the cylindrical cell—positions the cell-division machinery. The purification and reconstitution of the Min system in vitro showed that only two proteins, MinD and MinE, a lipid bilayer that mimics the cell membrane and ATP as chemical fuel are required for the formation of patterns[3]. This reconstitution provides a minimal system that enables precise control of reaction parameters and geometrical constraints, thus making the Min system a paradigmatic model system for protein-based pattern formation[4–7]. A remarkably rich plethora of dynamic membrane-bound patterns is found in vitro; predominantly traveling waves and spirals[3], but also "mushrooms", "snakes", "amoebas" and "bursts"[8,9], chaotic patterns[10,11], "homogeneous pulsing"[12–14] as well as quasi-stationary labyrinths, spots, and mesh-like patterns[11,15]. Critically, these patterns differ qualitatively and quantitatively from the pole-to-pole oscillations observed in vivo. This dichotomy has remained puzzling so far. It raises the central question whether the in vitro and in vivo patterns even share the same underlying pattern-forming mechanism, and, more generally, how we can gain insights on in vivo self-organization from in vitro studies with reconstituted proteins.

In general, protein patterns form by the self-organized interplay of chemical reactions and diffusive transport. The diffusive transport is caused by cytosolic (bulk) concentration gradients that form due to the attachment and detachment of proteins to and from the membrane surface. In turn, local changes in total protein concentrations change the reaction kinetics. Mathematically, this interplay is described in terms of mass-transport modes that can be identified by means of a linear stability analysis[5,16,17]. This theory has been recently developed as an extension of the canonical theory for Turing pattern formation in reaction-diffusion systems[18].

In *E. coli* cells, the Min proteins are cyclically transported from one cell pole to the other and back, giving rise to so-called pole-to-pole oscillations. In the underlying lateral mass-transport mode, the cytosol acts as a "transport medium" for transport along the length of the cell[17,19]. In contrast to the cellular confinement of an *E. coli* cell, typical in vitro setups have a much larger bulk volume per membrane area. Therefore, significant concentration gradients form not only laterally along the membrane, but also in the direction orthogonal to the membrane. These vertical gradients have been shown in a recent theoretical study to facilitate oscillations driven by diffusive transport of proteins between the membrane and the bulk further away from the membrane[5]. In these membrane-to-bulk oscillations, the bulk acts not only as a transport medium but also as an effective particle reservoir. In experiments, membrane-to-bulk oscillations manifest as homogeneous "blinking" of vesicles, where the proteins oscillate between the membrane and the bulk without breaking the rotational symmetry of the spherical vesicle[12–14]. The presence of at least two fundamentally different mass-transport modes—lateral and vertical—that may drive pattern formation in the Min system raises two important questions: What are the key control parameters that determine the operation of these modes? Which of these modes govern pattern formation in vivo, and which govern pattern formation in the classical in vitro setups? Answering these questions will be crucial to identify mechanistic similarities and differences between the well studied Min-pattern-formation phenomena in vivo and in vitro, and help unify our understanding of this remarkable dynamical system.

Here, we aim to answer these questions, by combining in vitro experiments with numerical simulations and the theoretical analysis of an established mathematical Minmodel. Importantly, the experiments are performed in a setup that allows us to unambiguously identify and distinguish the mass-transport modes driving the observed dynamics. The key role of bulk-concentration gradients in the different mass-transport modes suggests that the geometric confinement of the bulk volume, quantified by the ratio of bulk volume to surface area (short: bulk-surface ratio) is a key parameter that controls pattern formation. To systematically study the role of this parameter, we use laterally large microchambers with flat surfaces on top and bottom and carefully control the heights of these chambers in the range between 2 and 60 μm (see Fig. 1A). The microchambers' height directly determines the bulk-surface ratio while keeping all other parameters—like the lateral dimensions of the system, total protein concentrations, and kinetic rates—fixed. Specifically, the microchamber height confines only the vertical concentration gradients, while leaving the lateral mode unaffected such that membrane-bound protein patterns can evolve freely in the lateral directions along the membrane surfaces. This eliminates the confounding effects of lateral confinement that is inherent to previous experimental setups using various 3D confinements[8,12–14,20–22]. The second advantage of our microchambers is that we can directly study the correlation (synchronization) between the membrane-bound patterns at the two juxtaposed membrane surfaces. This correlation reveals the coupling between the two membranes through the bulk in-between them and provides evidence for the vertical bulk gradients. In particular, we find a vertical mass-transport mode that is specific to the two-membrane setup, where it drives membrane-to-membrane oscillations. This mode becomes unstable only above a critical bulk height and marks the transition to a new dynamical regime. For lower bulk heights, only the lateral mass-transport mode is unstable, corresponding to the situation in cells and representing the mechanistic analogue to the in vivo system. Above a critical bulk height, our theory predicts that multiple different mass-transport modes become unstable leading to multistability of patterns, i.e., the emergence of different patterns under the same set of conditions. We confirm this prediction experimentally and compile an experimental phase diagram in the parameter plane of bulk height and MinE to MinD concentration ratio. The topology of this phase diagram agrees with the prediction from linear stability analysis and numerical simulations.

Taken together, systematic variation of the bulk height experimentally confirms our theoretical prediction that the bulk-surface ratio is the key parameter that continuously connects the in vivo and in vitro. The (qualitative) in vivo vs in vitro dichotomy is resolved by our finding that they are governed by different pattern-forming mechanisms. The key geometric constraint that gives rise to the in vivo phenomenology (pole-to-pole oscillations, stripe oscillations in long cells, absence of homogeneous oscillations/"blinking") is the vertical (radial) confinement of the cytosolic volume, not the lateral confinement.

## Results

**Finite bulk heights lead to drastically different Min patterns.** To study the effect of the bulk-surface ratio on Min-pattern formation in vitro, we need to control this parameter without imposing lateral spatial constraints that affect pattern formation. To achieve this, we created a set of PDMS-based microfluidic chambers of large lateral dimensions (2 × 6 mm) but with a low height in a range from 2 to 57 μm (Fig. 1A). In these wide chambers, the membrane-bound protein patterns can freely evolve in the lateral direction while we study the effects of vertical bulk-concentration gradients, which are constrained by the microchamber height. All inner surfaces of the microchambers

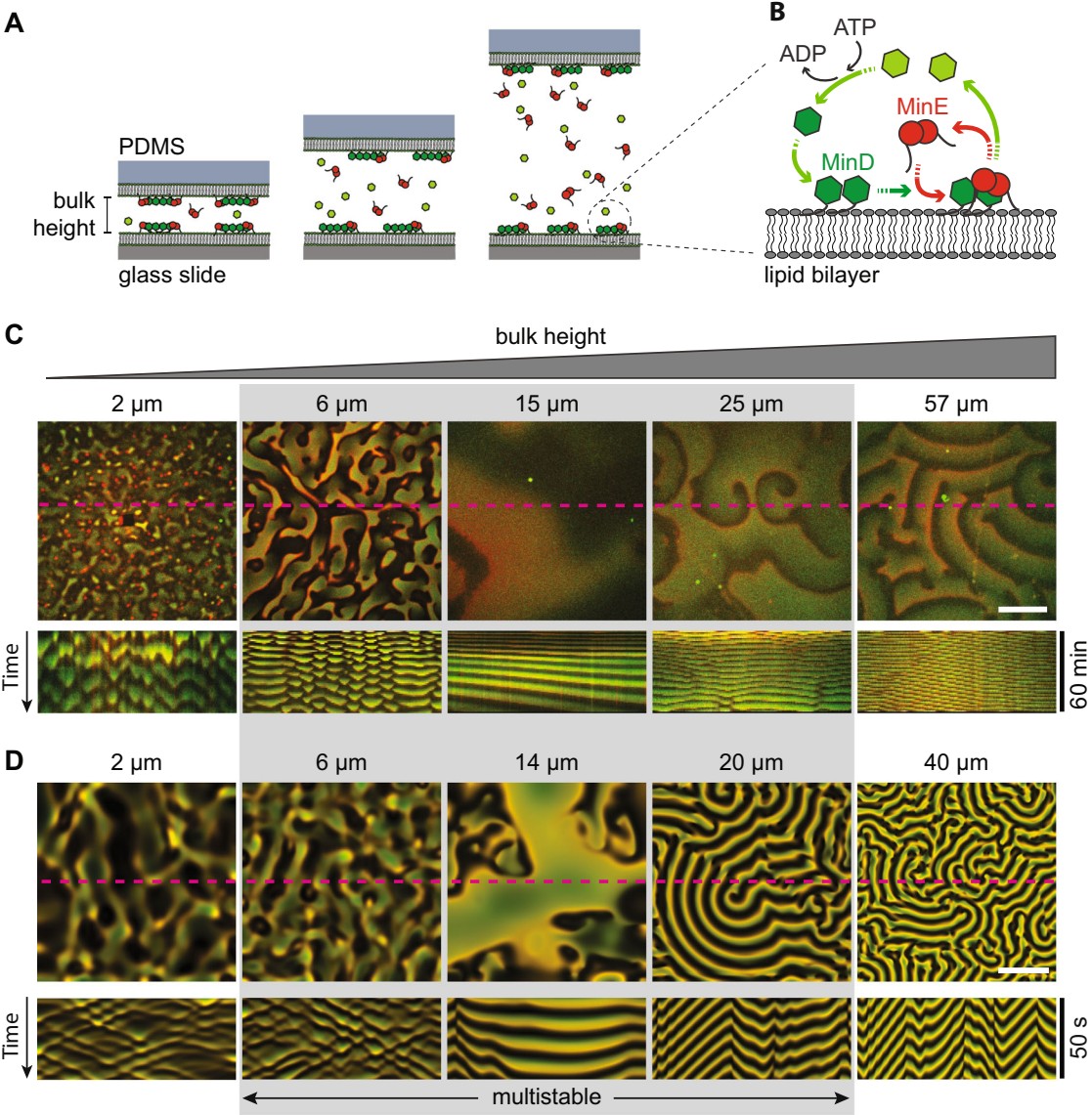

**Fig. 1 Effect of a change of a bulk height on Min-pattern formation. A** General concept of the experimental setup. MinD and MinE proteins were reconstituted in laterally large and flat microchambers of different heights. All inner walls of microchambers were covered with supported lipid bilayers made of DOPC:DOPG:TFCL (66 : 32.99 : 0.01 mol%) mimicking the *E. coli* membrane composition. Min proteins cycle between bulk and the membrane upon which they self-organize into dynamic spatial protein-density patterns. **B** Min-protein interaction scheme. MinD monomers (light-green hexagons) bind ATP resulting in dimerization and cooperative accumulation on membrane (dark-green hexagons). Next, MinE dimers bind to MinD, activating its ATPase activity, detachment from the membrane, and diffusion to the bulk where ADP is exchanged to ATP, and the cycle repeats. **C** Influence of the bulk height on Min-pattern formation. Snapshots show overlays of MinD channel (green) and MinE channel (red) imaged on the membrane 30 min after injection. Kymographs below were generated along the dashed magenta lines. In each microchamber, the concentrations of the reconstituted proteins are 1 µM MinE and 1 µM MinD (corresponding to a 1:1 *E:D* ratio). The gray shaded area marks the range of bulk heights where different patterns are observed in repeated experiments (multistability), cf. Fig. 3. **D** Snapshots and kymographs from numerical simulations of the reaction-diffusion model describing the skeleton Min-model in a three-dimensional box geometry with a membrane on top and bottom surfaces and reflective boundaries at the sides (see Supplementary Methods 1 and 3 for details). The colors are an overlay of MinD density (green) and MinE density (red) on the top membrane. Parameters (from left to right): $H = 2\,\mu m$, $E:D = 0.8$; $H = 6\,\mu m$, $E:D = 0.75$; $H = 14\,\mu m$, $E:D = 0.75$; $H = 20\,\mu m$, $E:D = 0.725$; $H = 40\,\mu m$, $E:D = 0.625$. Lateral system dimensions: $200 \times 200\,\mu m$. The remaining, fixed model-parameters are given in Supplementary Table 1. (White scale bars in **C** and **D**: 50 µm).

were covered with supported lipid bilayers composed of DOPG: DOPC (3:7), which has been shown to mimic the natural *E. coli* membrane composition[23]. Proteins were administered by rapid injection of a solution containing 1 µM of MinE and 1 µM of MinD proteins, together with 5 mM ATP and an ATP-regeneration system[21].

Figure 1C and Supplementary Movie 1 show snapshots and kymographs of the characteristic patterns observed in microchambers of different heights. We clearly observe distinct Min

patterns that can be identified qualitatively by simple visual inspection: standing wave chaos, homogeneous oscillations, and traveling (spiral) waves. Moreover, for intermediate bulk heights (range shaded in gray in Fig. 1) we observe that the system has the capacity to robustly exhibit different pattern types for the same conditions (parameters). This *multistability* is discussed further below.

For low bulk heights (2–6 µm in Fig. 1C) we observe incoherent wave fronts of the protein density propagating from

low density towards high density regions, thus continually shifting these regions in a chaotic manner as can be seen in the kymographs. We will refer to these patterns as standing wave chaos. The chaos-like character is also evidenced by the irregular shapes and nonuniform propagation velocities of wave fronts within the same pattern (see Supplementary Fig. 1). Still, these patterns clearly have a characteristic wavelength.

For an intermediate bulk height (15 µm in Fig. 1C), we observe patterns with large areas that have fairly homogeneous Min-protein density and temporally oscillate as a whole. We refer to these patterns as *homogeneous oscillations*. (We will use this term whenever there are large patches where the temporal oscillations are in-phase, yielding a spatially homogeneous pattern in these patches.) Phenomenologically similar oscillations have been observed as initial transients in some previous experiments[8,24]. In contrast, however, the oscillations that we observed for intermediate bulk heights persisted throughout the entire duration of the experiment (90 min).

In the large bulk-height regime (57 µm in Fig. 1C) we find traveling waves that are characterized by high spatial coherence of the consecutive wave fronts that propagate together at the same velocity and with a well-controlled wavelength. Finally, the wave patterns found at 25 µm shows phenomena indicative of defect-mediated turbulence: continual creation, annihilation, and movement of spiral defects (Supplementary Movie 2). This behavior is commonly found in oscillatory media at the transition between spiral waves and homogeneous oscillations/phase waves[25,26].

Taken together, we find that the bulk height has a profound effect on the phenomenology of Min-protein pattern formation. Notably, the bulk-surface ratio at the lowest bulk height (2 µm) is of the same order of magnitude as in *E. coli* cells, which have a diameter of about 0.5–1 µm. However, there is no obvious phenomenological correspondence between the in vivo system and the laterally unconfined in vitro system as the patterns found in these two settings differ qualitatively. Despite this lack of obvious phenomenological correspondence, the theoretical analysis carried out below, based on a minimal model of the Min-protein interactions, allows us to find the connection between the in vivo and in vitro dynamics by identifying the underlying mesoscopic mechanisms (mass-transport modes).

**A minimal model reproduces the salient, qualitative pattern features**. To explain the observed diversity of patterns found in experiments, we performed numerical simulations and a theoretical analysis. We used a minimal model of the Min-protein dynamics that is based on the known biochemical interactions between MinD and MinE (Fig. 1B). This model encapsulates the core features of the Min system and has successfully reproduced and predicted experimental findings in a broad range of conditions both in vivo[19,27,28] and in vitro[5,10]. Finite-element simulations of this model in the same geometry as the microchambers (laterally wide cuboid with membrane on both top and bottom surfaces, see Supplementary Fig. 2) qualitatively reproduce three pattern types found in experiments in the three regimes of bulk heights, as shown in Fig. 1D and Supplementary Movie 3. (A fourth type of pattern, termed "amoebas" is found in experiments outside the parameter regime accounted for by our minimal model; see below).

For low bulk heights (0.5–5 µm in Fig. 1D), the model exhibits standing wave chaos (incoherent fronts that chaotically shift high- and low-density regions) in close qualitative resemblance of the patterns found experimentally. For intermediate bulk heights (5–15 µm in Fig. 1D), we find nearly homogeneous oscillations, meaning large areas with a nearly homogeneous protein density that oscillate temporally. Gradients in the oscillation phase lead to

the impression of propagating fronts, with a velocity inversely proportional to the phase gradient (sometimes called "pseudo waves" or "phase waves" in the theoretical literature[29,30]). In contrast to genuine traveling waves, phase waves are merely phase-shifted local oscillations. They do not require lateral material transport (lateral mass redistribution) and the visual impression of "propagation" is merely a consequence of the phase gradient. In addition, continual creation and annihilation of topological defects in the phase (like the two spiral defects in Fig. 1C, 25 µm) can give rise to defect-mediated turbulence[25,31]. We note that the pattern observed for 14 µm possess the visual characteristics of such turbulence. However, a detailed quantification of this phenomenon is beyond the scope of the present work. Finally, for large bulk heights (>20 µm in Fig. 1D), we find traveling (spiral) waves. This is in agreement with simulations performed for the same reaction kinetics in a setup with a planar membrane on only one side of a large bulk volume[5].

In summary, the model qualitatively reproduces the salient features of our experimental observations across the whole range of bulk heights remarkably well (Fig. 1C, D). However, the characteristic wavelength and oscillation periods of the patterns are not matched quantitatively (although they are of the same order of magnitude). Given the lack of a theoretical understanding of the principles underlying nonlinear wavelength selection, the large number of experimentally unknown kinetic rates, and the potential need to further extend the model[10,11], fitting parameters is beyond the scope of the present study (please refer to the Discussion and Supplementary Discussion 2 for further elaboration on the question of length- and timescales).

Rather than wavelength selection, our focus here are the fundamental pattern-forming mechanisms, which can be identified by robust, qualitative signatures of these mechanisms: the topology of the phase diagram in the parameter space of bulk height and MinE/MinD concentration ratio, and the synchronization of patterns between the two opposite membrane surfaces.

**Distinct mass-transport modes underlie pattern formation at different bulk heights**. To identify the pattern-forming mechanisms governing the dynamics for different bulk heights, we performed a linear stability analysis of the homogeneous steady states. This analysis identifies the mass-transport modes that govern pattern formation (see Supplementary Method 2 for technical details). For each set of parameters (such as total protein concentrations, bulk height and kinetic rates), only those modes that are *unstable* contribute to the dynamics, while the stable modes are "inactive".

We find three types of mass-transport modes: a lateral mode, transporting mass along membrane, and two types of vertical modes transporting mass orthogonally to the membranes (membrane-to-membrane and membrane-to-bulk). The vertical modes do not require lateral coupling (lateral redistribution of mass) and therefore correspond to local instabilities a recently developed theoretical framework for pattern formation[5,16]. Here, we use the term local regarding the direction along the membrane, as an antonym to lateral. Importantly, the local instabilities still involve spatial gradients in the direction *orthogonal* to the membrane, i.e., in the vertical direction in the microchamber geometry.

The phase diagram in Fig. 2A shows the regimes where the three mass-transport modes are unstable as a function of the bulk height and the ratio of MinE concentration and MinD concentration (*E:D* ratio for brevity). Notably these regimes largely overlap, meaning that multiple distinct mass-transport modes can contribute to the dynamics at the same time.

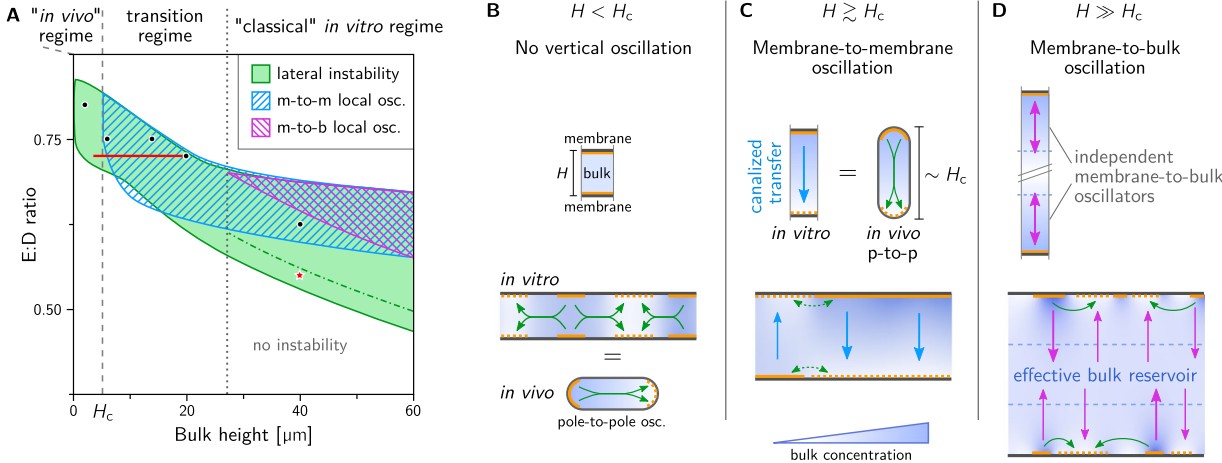

**Fig. 2 Distinct lateral and vertical mass-transport modes at different bulk heights. A** Phase diagram for bulk height and $E{:}D$ ratio showing three types of mass-transport modes, identified by linear stability analysis, that exist in overlapping regimes: lateral instability (green), local membrane-to-membrane instability ("m-to-m", blue) and local membrane-to-bulk instability ("m-to-b", magenta). See Supplementary Fig. 3 for representative dispersion relations in the various regimes. Green dot-dashed line marking the transition from chaotic to coherent patterns[5]. A representative example of a chaotic pattern is shown in Supplementary Movie 4 for the parameter combination marked by the red star. Black dots mark the parameters used for the simulations shown in Fig. 1d. Red line: parameter range for adiabatic sweeps shown in Supplementary Fig. 6 demonstrating hysteresis as a signature of multistability. Panels **B**–**D** illustrate the mass-transport modes at different bulk heights. The top row shows laterally isolated compartments to illustrate local vertical oscillations due to vertical bulk gradients. The bottom row illustrates the interplay of lateral and vertical mass-transport modes in a laterally extended system. **B** For low bulk height, the bulk height is too small for significant vertical concentration gradients to form. Hence, a laterally isolated compartment does not exhibit any instabilities (top). In a laterally extended system (bottom), exchange of mass can drive a lateral instability (green arrows); see Supplementary Movie 5. The cartoon of an *E. coli* cell illustrates that this instability also underlies pattern formation in vivo; see Supplementary Movies 19 and 20. **C** For bulk heights above $H_c$, vertical concentration gradients become significant enough to enable vertical membrane-to-membrane oscillations (blue arrows); see Supplementary Movie 6. These oscillations do not require lateral exchange of mass, i.e., they occur in a laterally isolated bulk column (top). The cartoon of an *E. coli* cell illustrates that the membrane-to-membrane oscillations in a laterally isolated compartment can also be pictured as equivalent to in vivo pole-to-pole oscillations. The laterally extended in vitro system constitutes a continuum of such oscillators (bottom). **D** For bulk heights larger than the penetration depth of vertical gradients, the top and bottom membrane effectively decouple (see Supplementary Movies 7 and 8). In this regime, which corresponds to the classical in vitro regime, the bulk in-between the membranes acts as an effective protein reservoir that facilitates membrane-to-bulk oscillations.

**Low bulk height: only lateral oscillations**. At low bulk height, diffusion mixes the bulk in vertical direction quickly, such that no substantial vertical protein gradients can form (see Supplementary Movie 5). Consequentially, the vertical mass-transport modes are stable. In other words, the up-down symmetry of the system is not spontaneously broken. This corresponds to the azimuthal symmetry of rod-shaped cells, which is not broken by pole-to-pole oscillations. If a compartment were isolated laterally, it would not exhibit any instabilities, because the lateral isolation suppresses the lateral mode (see Fig. 2B, top). In the laterally extended system, there is an instability driven by lateral mass redistribution (illustrated by the green arrows in Fig. 2B, bottom) due to lateral cytosolic gradients[16]. Lateral mass redistribution also underlies the Turing instability and in vivo Min patterns[19]. Consistently, simulations performed in a cell geometry with the dimensions of an *E. coli* bacterium, using the same kinetic rates as in the remainder of this study (see Supplementary Table 1), show pole-to-pole in normal sized cells and stripe oscillations in long (filamentous) cells, in agreement with experimental observations (see Supplementary Movie 19). For a detailed analysis of the in vivo dynamics see ref. [32]. If, instead of lateral mass transport, in vivo Min-protein patterns were driven by vertical oscillation (mass transport) modes, spherical minicells would blink homogeneously. Homogeneous blinking of cells has not been observed experimentally[33,34]. We conclude that the patterns observed in vitro for low bulk height are governed by the same mechanism as that in vivo—a lateral mass-transport mode (corresponding to a "Turing instability"). This finding reveals the underlying cause for the correlation between bulk depletion and the occurrence of

standing waves (pole-to-pole/stripe oscillations in cells and "bursts" in vitro) reported in ref. [9]: bulk depletion by attachment of proteins to the membrane generates the lateral gradients that drive the standing wave pattern.

**Intermediate bulk height: membrane-to-membrane oscillations**. For a sufficiently large bulk height, vertical concentration gradients drive a membrane-to-membrane mass-transport mode as illustrated in Fig. 2C (cf. Supplementary Movie 6). We define the critical height $H_c$ as the lowest bulk height for which this mode becomes unstable, i.e., starts to contribute to the dynamics. It is determined by the penetration depth of vertical concentration gradients in the bulk that arise as a consequence of the attachment and detachment of proteins at the membranes. The value of $H_c$ depends on the kinetic rates and bulk diffusivities. For the parameters used here, it is approximately 5 μm (see Fig. 2A). Characteristically, these membrane-to-membrane oscillations are in antiphase between the top and the bottom membrane. This alternation in protein density will later serve as a signature of the vertical concentration gradients that drive membrane-to-membrane oscillations in the experiment.

Notably, an analogy can be drawn to in vivo pole-to-pole oscillations. The two-membrane "patches" at the top and bottom of a laterally isolated ("local") compartment in the in vitro system can be pictured as analogous to the cell poles in vivo. Hence, the vertical membrane-to-membrane oscillations in vitro share their mechanism of operation with the in vivo pole-to-pole oscillations. Yet there are two key differences. First, the orientation of the

gradient is turned by 90 degrees (compare Fig. 2B bottom with Fig. 2C top). Second, the extended membrane surfaces in the in vitro system constitute a continuum of laterally coupled oscillators (i.e., an oscillatory medium, see Fig. 2C bottom) while the cellular confinement accommodates only a single oscillator (see Fig. 2B top). Therefore, even though some aspects of the in vivo system are present for bulk heights above $H_c$, we denote only the regime $H < H_c$ as "in vivo like" (see Fig. 2A).

**Large bulk height: membrane-to-bulk oscillations.** When the bulk height is larger than the penetration depth of the bulk gradients, the bulk further away from the membranes acts as a protein reservoir and facilitates oscillations between the membrane and the bulk reservoir—individually and independently for both the top and bottom membrane, as illustrated by magenta arrows in Fig. 2D (see also Supplementary Movies 7 and 8). Diffusion from the membrane to the bulk reservoir and back provides the delay that underlies these membrane-to-bulk oscillations for a large bulk height. The bulk reservoir in-between the membranes acts as a buffer that decouples the two membranes. Thus, for large chamber heights, the microchamber geometry with two membranes is equivalent to two independent systems with a single membrane, such as "classical" in vitro setups with a large bulk volume on top of a supported lipid bilayer.

**Fully developed nonlinear patterns and multistability.** The different patterns shown in Fig. 1D correspond to the three mass-transport modes discussed above: Lateral mass transport alone drive standing waves (Supplementary Movie 5). Dominance of vertical membrane-to-membrane mass transport leads to laterally homogeneous oscillations (Supplementary Movie 6). These homogeneous oscillations clearly demarcate the transition from the in-vivo-like regime, where only lateral oscillations but no vertical oscillations exist, to the in vitro regime where vertical oscillations come into play. Notably, in the "transition regime" lateral instability and vertical membrane-to-membrane instability coexist in largely overlapping regions of parameter space (Fig. 2A). As a result of the competition between these distinct mass-transport modes, we expect multistability, that is, the emergence of different pattern types for the same set of conditions, depending on the initial conditions (see Supplementary Fig. 5). Moreover, in simulations where the bulk height was increased/decreased very slowly compared to the oscillation period of patterns show pronounced hysteresis in the transitions between the different pattern types (Supplementary Fig. 6; simulations details given in Supplementary Method 4). Moreover, for intermediate bulk height (~18 μm) we observe spatiotemporal intermittency (Supplementary Fig. 7). This phenomenon can be pictured as the coexistence of homogeneous oscillations, traveling waves, and standing waves in space where they continually transitioned from one to another over time.

For large bulk height, the interplay of lateral oscillations and vertical membrane-to-bulk oscillations drives traveling waves (Supplementary Movie 7) and standing wave chaos at low $E{:}D$ ratios (see Supplementary Movies 4 and 8 for simulations the full geometry (2 + 3D) and in slice geometry (1 + 2D), respectively). The large bulk-height regime was investigated in detail in a previous theoretical study[5], which focused on the transition from standing wave chaos to traveling waves.

In passing, we note that the patterns we find in numerical simulations have large amplitude. It is no a priori clear whether the linear stability analysis of the homogeneous steady state is informative regarding such large-amplitude patterns. In In Supplementary Method 3, we briefly describe how a recently developed theoretical framework called "local equilibria theory"

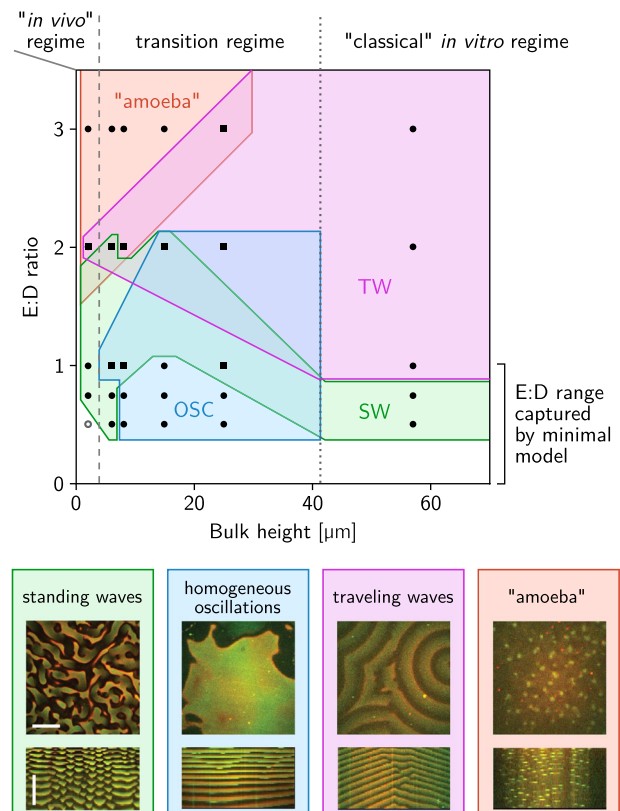

**Fig. 3 Experimental phase diagram showing multistability.** Black symbols mark parameter combinations where experiments were performed. To vary the E:D ratio, the concentration of MinE was varied from 0.5 to 3 μM at a constant MinD concentration of 1 μM. Circles (squares) mark monostable (multisatble) regions, where one (multiple) pattern types were found in repeated experiments. The observed pattern types are indicated by the colored regions; the legend shows representative snapshots and kymographs (see Supplementary Fig. 9 for representative snapshots for each parameter combination and pattern type separately). Note that the topology of the phase diagram agrees with the prediction from linear stability analysis (Fig. 2a) within the regime captured by the minimal model ($E{:}D < 1$). (Scale bars: 50 μm and 30 min).

can be used to characterize large-amplitude patterns locally and regionally[5,16]. Centrally, this framework utilizes the fact that the Min-protein dynamics conserve the total amounts of MinD and MinE. We performed an analysis using local equilibria theory for the patterns found in numerical simulations to confirm the intuition from linear stability analysis. In particular, this analysis shows explicitly how the different mass-transport modes described above drive the pattern of fully developed patterns that have large amplitude (see In Supplementary Method 3 and Supplementary Movies 16–18).

To experimentally test the predicted multistability, we systematically varied the bulk height (from 2 to 57 μm) and the $E{:}D$ ratio (from 0.5 to 3), and repeated the experiment several times ($N = 2$–5) for each parameter combination. Figure 3 shows the phase diagram obtained from this large assay. Detailed descriptions and quantifications the patterns observed in the different regimes of the phase diagram are provided in Supplementary Discussion 1. Critically, within the regime captured by the minimal model ($E{:}D < 1$), the topology of this experimental phase diagram agrees with the prediction from linear stability analysis (Fig. 2A). In particular, for intermediate bulk height, we observed qualitatively different patterns in repeated experiments with the same parameters, which clearly indicates multistability (see the overlapping regions in Fig. 3, and

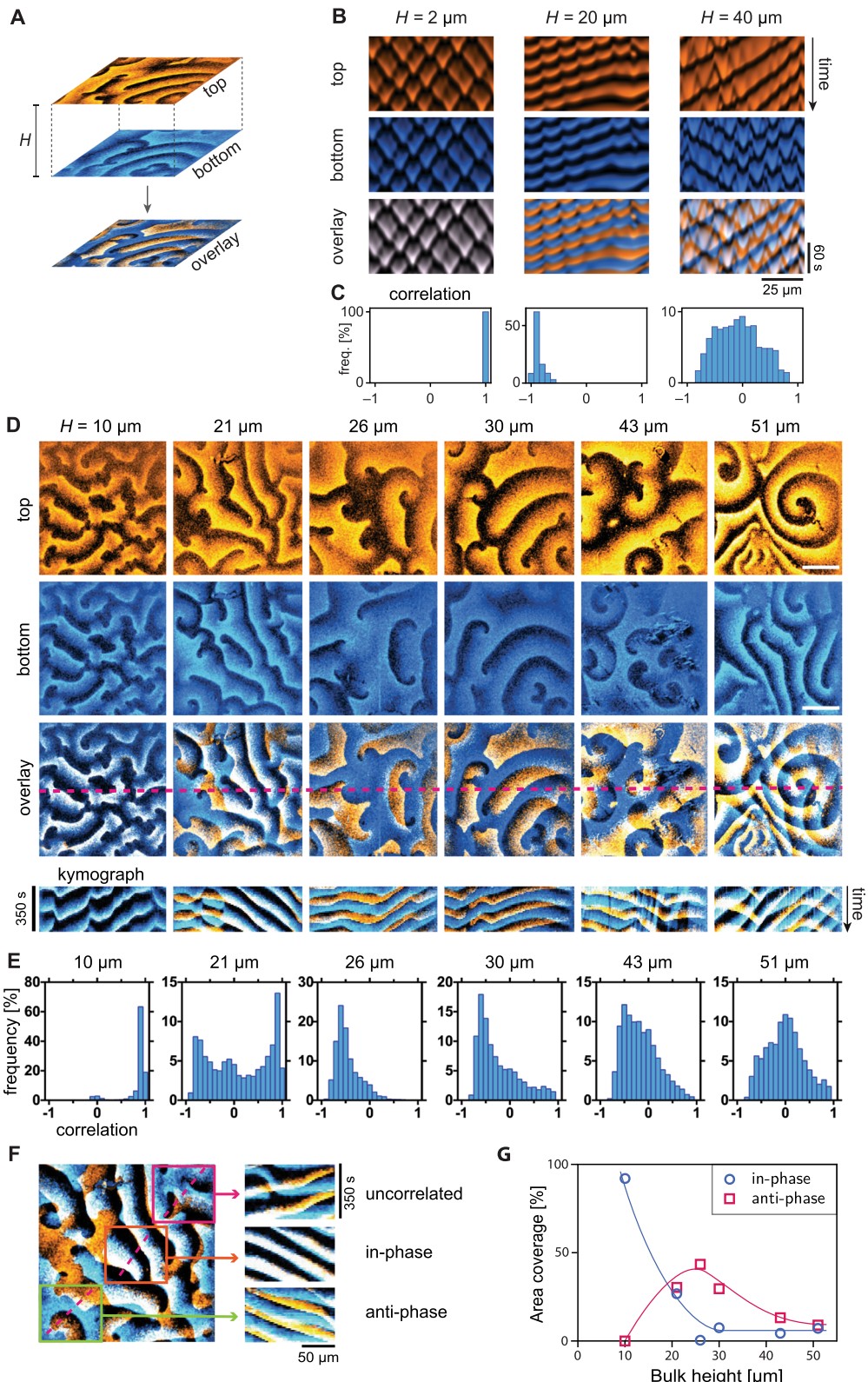

Supplementary Movie 15 showing an example of threefold multi-stability). By contrast, for low bulk height (2 μm) we found only a single instance of (twofold) multistability ($H = 2\,\mu m$, $E{:}D = 2$), whereas for large bulk height (57 μm) we did not observe any multistability at all. In agreement to these experimental findings, numerical simulations for small and large bulk heights do not show multistability of qualitatively different patterns. Note that for $E{:}D > 1$,

the minimal model does not exhibit pattern formation[19]. An extension to the model, accounting for the switching of MinE between an active and an inactive conformation, is required to capture pattern formation in this regime[10]. This extension, however, would significantly increase the number of parameters and variables and require a readjustment of parameters to the observed phenomenology at the computational cost of several weeks CPU

**Fig. 4 Min crosstalk between opposite membranes. A** Patterns form on the membranes both on the top and bottom surface of the microchambers. Overlaying these patterns in different colors (blue and orange) reveals the synchronization between them. In the overlay, blue and orange additively mix to white such that white areas signify high concentration on both membranes. Black indicates areas of coinciding low protein density. **B** Kymographs from simulations in slice geometry (1 + 2D) showing perfect in-phase synchrony of patterns at low bulk height, antiphase synchrony driven by membrane-to-membrane mass transport for intermediate bulk height and desynchronization for large bulk height, where two membranes effectively decouple. (E:D ratios from left to right: 0.75, 0.725, and 0.55). **C** Histograms of the correlation between top and bottom membrane corresponding to the kymographs in **B**. The correlation was calculated between concentration time-traces over a 100 s interval at regularly spaced spatial positions ($\Delta x = 0.25\,\mu m$). **D** Snapshots and kymographs from simultaneous (<0.1 s delay) imaging of MinE on the top (orange) and bottom (blue) membrane in microchambers of different heights (1 µM MinD, 1 µM MinE). Bars correspond to 50 µm. **E** Each field of view (FOV) was divided into a grid of cells for which the correlation analysis was performed individually. Histograms show frequency distribution of correlations of individual cells in the grid measured for 30 timepoints in each FOV. Perfect in-phase correlation corresponds to a correlation value of 1 and perfect antiphase to a value of −1, respectively; lack of correlation corresponds to a correlation measure of 0. **F** Example of coexistence of in-phase and antiphase synchrony within adjacent spatial regions. **G** Classification of top-bottom correlation as a function of bulk height, extracted from the histograms in panel **D**. Correlation values above 0.7 are classified as correlated, values less than −0.3 as anticorrelated. (Source data are provided as a Source Data file).

time. Given that the minimal model captures the phenomenology we are interested in, we refrain from considering the model extension explicitly.

**Interplanar pattern synchronization reveals vertical mass-transport modes in experiments.** Recall that the vertical synchronization of patterns on the opposite (top and bottom) membranes is a key signature that distinguishes the lateral mass-transport mode for low bulk height, the vertical membrane-to-membrane mass-transport mode for intermediate bulk height, and the vertical membrane-to-bulk mass-transport mode for large bulk height. Respectively, these modes cause strong in-phase coupling, antiphase coupling, and de-coupling of the dynamics on the two opposites membranes (Fig. 4B).

In the following, we will use these characteristics to infer the underlying mass-transport modes directly from the experimentally observed patterns. To analyze interplanar synchronization of patterns, we imaged the Min patterns on both membranes simultaneously (delay <0.1 s) in a set of experiments with bulk heights ranging from 10 to 51 µm at 1:1 *E:D* ratio. Figure 4D and Supplementary Movie 9 show the patterns found in this set of experiments. We calculated correlation histograms to quantify the observed synchronization between the membrane-bound patterns forming on the two-membrane surfaces (Fig. 4E). In agreement with the predictions, we find fully synchronized patterns for low bulk height (10 µm). For intermediate bulk heights (21–30 µm), we find coexisting spatial regions where patterns are clearly synchronized in antiphase as well as regions of in-phase synchronization. In particular, strong anticorrelation, indicating antiphase synchronisation, is found for bulk heights 26 µm and 30 µm. This is a strong indication of the membrane-to-membrane oscillations predicted by the theoretical model. Notably, coexistence of in-phase synchronized patterns and antiphase oscillations in neighboring regions of the membranes is also observed in numerical simulations in three-dimensional geometry (Supplementary Fig. 7 and Supplementary Movie 10). Finally, and in agreement with the theoretical prediction, patterns become increasingly desynchronized between the opposite membranes for large bulk heights, with a near-complete dissimilarity for (51 µm).

Figure 4G summarizes the bulk height dependency of the pattern correlation and clearly shows that in-phase correlation is maximal at small bulk heights, anticorrelation peaks for intermediate bulk heights, and that the overall correlation decreases as bulk height increases and both membranes decouple from each other. These findings provide strong experimental evidence for the existence and importance of vertical concentration gradients in the bulk. Critically, our theoretical analysis above has shown that the synchronization across the bulk is a consequence of the different mass-transport modes underlying pattern formation and hence reveals the role of these mass-transport modes in the experimental system.

## Discussion

The starting point for this study was the puzzling qualitative and quantitative differences between the phenomena exhibited by the MinDE-protein system in vivo and in vitro. We found that different patterns emerge from and are maintained by distinct pattern-forming mechanisms (mass-transport modes) that depend on how far concentration gradients can penetrate into the cytosolic bulk, and thus on the geometry of the system, especially the bulk height. Thus, a principled theoretical approach together with a minimal model and direct experimental verification has enabled us to disentangle the Min system's complex phenomenology and identify the pattern-forming mode in vitro that is mechanistically equivalent to the mode driving in vivo patterns. This implies that no new biochemical insight is necessary to qualitatively resolve the dichotomy between in vivo and in vitro phenomenology.

Rather, our results show that the bulk height, or more generally the bulk-surface ratio, is a control parameter of equal importance as the protein concentrations and kinetic rates. Importantly, this implies that to faithfully emulate the in vivo system in an in vitro setup, one needs to confine the bulk volume sufficiently to suppress vertical mass-transport modes. The presence of homogeneous oscillations (sometimes called "blinking" or "pulsing") of Min proteins in vesicles and micro-droplets with diameters as small as 10 µm[12–14] indicates that previous in vitro setups have not succeeded in suppressing the vertical mass-transport modes. Conversely, we predict that *E. coli* cells sculpted into spherical shape of sufficient diameters (about 10 µm) will exhibit homogeneous oscillations.

A major phenomenological feature of patterns that is typically discussed in the context of the in vivo vs in vitro dichotomy is the pattern wavelength, which is ~50 µm in vitro compared to ~5 µm in vivo. Unfortunately, the principles of wavelength selection of highly nonlinear patterns in general, and of the Min-protein patterns in particular, remains an open problem, both theoretically and experimentally. In the Supplementary Discussion 2, we provide a more in-depth discussion of the topic of length-scale selection and show that the width of interfaces of the large-amplitude patterns, rather than their wavelength, is characteristic for the underlying pattern-forming mode. Future studies, pursuing a systematic understanding of wavelength selection, might ultimately answer why the Min-pattern wavelengths are so different in vivo and in vitro. Rather than attempting a quantitative fit, we relied on robust qualitative features to relate our theoretical results to experimental observations. These qualitative features,

predicted from the model and confirmed in experiments, are the topology of the phase diagram (critically including a region of multistability) and the synchronization of patterns between the two-membrane surfaces. Remarkably, by simply varying the distance between the membrane surfaces (i.e., the bulk height) we found both in-phase and antiphase synchronization. These interesting phenomena deserve more detailed experimental and theoretical investigations.

From a broader perspective, bulk-surface coupling (cycling of proteins between membrane and cytosol) is a fundamental feature of protein-based pattern formation in cells[7,32]. Thus our results suggest that geometric properties, like the bulk-surface ratio, will have an impact on the dynamics in many such systems.

Studying parameters that can be varied on a continuous axis—namely the $E:D$ ratio and the microchamber height—has been key to our study. It has allowed us to obtain a "phase diagram", which reveals the transitions and relations between the regimes where different pattern-formation modes are operating. This lays the foundation to tailor the Min system for specific applications[35].

## Methods
Experiments were performed with purified Min proteins in PDMS microchambers and glass flow cells coated with lipid bilayers as described below.

The mathematical model accounting for the core set of Min-protein interactions[5,19,27], termed skeleton Min-model, was analyzed using linear stability analysis and numerical simulations as described in Supplementary Methods 1–4.

**Chemicals**. All phospholipids used in this study (1,2-dioleoyl-sn-glycero-3-phos-phocholine; 1,2-dioleoyl-sn-glycero-3-phospho-(1'-rac-glycerol) (sodium salt); 1,1',2,2'-tetraoleoyl cardiolipin[4-(dipyrrometheneboron difluoride)butanoyl] (ammonium salt) TopFluor® Cardiolipin were purchased from Avanti Polar Lipids. Phosphoenolpyruvic acid was from Alfa Aesar. RTV 615 PDMS and crosslinker were purchased from Momentive. All other chemicals were purchased from Sigma–Aldrich (or otherwise as indicated).

**Microfabrication**. Fabrication of microstructures took place in a class 10000 cleanroom. Four-inch silicon wafers were cleaned with isopropanol and baked for 10 min at 200 °C. A thin layer of a primer hexamethyldisilazane (BASF) was spin-coated on the wafer at 1000 rpm for 1 min and baked at 200 °C for 2 min. Next, a thick layer of NEB22a resist (Sumitomo Chemicals) was deposited by spin-coating at 500 rpm for 15 s and prebaking at 110 °C for 3 min. Patterns were designed using the KLayout software and written into the NEB22a layer with the use of electron-beam lithography (EBPG-5000+, Raith GmbH) with a dose of 16 μC cm⁻², acceleration voltage of 100 kV, and with aperture 400 μm. Subsequently, the wafer was baked at 105 °C for 3 min. Non-cross-linked resist was removed using a MF322 developer (Dow Chemical Company) bath for 1 min followed by a subsequent 10% MF322 bath for 30 s and cleaning with distilled water for 30 s. Structures were etched with a Bosch deep reactive-ion etching process into the silicon wafer using inductive coupled plasma reactive-ion etcher (Adixen AMS 100 I-speeder). The etching step involved 200 sccm SF₆ for 7 s and the passivation step was done with 80 sccm C4F8 for 3 s. The etching time was chosen to achieve the desired height of microstructure. Resist was removed from wafer using oxygen plasma for 15 min. Quality of structures was examined using widefield microscopy and the height of structures was measured using a profilometer (Bruker Dektak XT). Wafers with microstructures were silanized over night using silane vapors in a vacuum chamber.

**Preparation of PDMS microchambers and glass flow cells**. A 5-mm layer of degassed PDMS (9:1 PDMS:curing agent ratio) was poured onto the wafer with microstructures and degassed in the vacuum chamber for an additional 1 h to remove any air bubbles. Afterwards, the PDMS was baked for 4 h at 80 °C and individual microchambers were cut out of a PDMS slab. Inlet and outlet holes were punched using a stainless steel 0.75 mm diameter biopsy punch (World Precision Instruments). Cover slips were cleaned in 1 M KOH for 1 h followed by 1 h methanol cleaning, both in a sonicator bath. Right before annealing, the PDMS device and a cover slip were briefly flushed with isopropanol, blown with an N₂ stream and treated with oxygen plasma for 20 s using oxygen plasma PREEN I (Plasmatic System, Inc.) with a flow of 1 SCFH of O₂. The PDMS device was then placed on the cover slip and baked for 10 min at 100 °C to facilitate bonding between PDMS and the glass. Right after that, a solution of small unilamellar vesicles (SUVs) containing 1 mg/ml lipids (including fluorescent lipids) in a buffer containing 25 mM Tris-HCl (pH 7.5), 150 mM KCl, and 5 mM MgCl₂ was incubated in the device at 37 °C for 30 min to facilitate the formation of supported lipid bilayer (SLB). Subsequently the device was washed for 10 min with the same buffer without lipids. The quality and cleanliness of the formation of membrane were

inspected using fluorescence microscopy. The chamber height was confirmed by acquiring z-stacks in multiple places over each device.

For the assay where we probe top-bottom synchronization, we used glass-based flow chambers. For their preparation, we used two rectangular cleaned cover slips (bottom one 22/50 mm, top one 5/30 mm) with thin parafilm stripes on both ends in between them as spacers. The top glass slip had two holes (inlet and outlet) drilled close to the opposite edges, to which tubing was attached. Sides were sealed using scotch tape to allow for elasticity. To form a SLB, the flow cell was filled with SUV solution through the inlet tube and incubated for 30 min. Subsequently, the flow cell was washed thoroughly using buffer to remove excess SUVs. Next, the distance between bottom and the top membrane was set using a custom-built screw-based mini press attached to the microscopic sample holder. The screw was pressing the top cover slip through a large metal pad to achieve a homogenous distance between cover slips over a a large area. The height between top and bottom membrane was monitored using fluorescent microscope by performing z-scan profile of membrane fluorophore emission. Upon reaching the desired height, flow cell was ready for injection of the proteins.

**Preparation of SUVs**. SUVs were prepared using a thin lipid hydration method. Lipids were dissolved in chloroform and evaporated in a glass vial under vacuum for 3 h. In all experiments the same lipid composition was used (DOPC:DOPG: TopFluorCardiolipin 67:33:0.02 mol%). Next, 25 mM Tris-HCl (pH 7.5), 150 mM KCl buffer was added to achieve a 5 mg/ml final lipid concentration, and a vial was incubated on a shaker for 1 h. Finally, lipid solution was extruded using Avanti Mini Extruder (Avanti Polar Lipids) through 30 nm filter. Aliquots were snap frozen in liquid nitrogen and stored in −80 °C. For preparation of the SLBs, an aliquot was thawed on ice, diluted in a buffer to a final composition 25 mM Tris-HCl (pH 7.5), 150 mM KCL 5 mM MgCl₂ and incubated in an ultrasonic bath at 37 °C for 15 min.

**Purification and labeling of Min proteins**. Min proteins were purified as described in ref. [21]. Briefly, 6×His-MinE and 6xHis-MinD were expressed in E. coli from the pET28a plasmid. Upon collection, cells were resuspended in lysis buffer containing 10 mM imidazole, 5 mM TCEP, complete protease inhibitor cocktail (Roche) and 100 μM ADP (for MinD only), and broken using a French Press. Cell debris free lysate was loaded on a HisTrap column (GE Helthcare). MinE was eluted using lysis buffer containing 250 mM imidazole and MinD 160 mM imidazole. Subsequently proteins were purified using size exclusion Sephacryl S-300 HR 16/60 column using buffer contining 50 mM Hepes pH 7.25 at 4 °C, 150 mM KCl, 10% v/v glycerol, 0.1 mM EDTA pH 7.4, and 80 μM of ADP (for the MinD protein). Protein concentration was measured using a QuantiProTM BCA assay kit (Sigma–Aldrich). MinD was labeled using NHS-Cy3 and MinE using Maleimide-Cy5 accordingly to the manufacturer procedure (GE Healthcare). The degree of labeling was Cy3-MinD 0:88, Cy5-MinE 0:45.

**Observation of Min patterns**. A solution containing 0.8 M MinD, 0.2 mM MinD-Cy3, 0.8 mM MinE, 0.2 mM MinE-Cy5, 5 mM ATP, 4 mM phosphoenolpyruvate, 0.01 mg/ml pyruvate kinase, 25 mM Tris-HCl (pH 7.5), 150 mM KCl, and 5 mM MgCl₂ was injected into the microchambers/flow cells using a syringe pump. The volume of protein solution was chosen 50 times larger than the volume of the microdevice to fully replace the buffer solution. To avoid accumulation of proteins on the membranes during filling the device, the solution was rapidly (5 s) flushed through. Observation of the patterns was initiated after a 30 min incubation period.

**Image acquisition and data analysis**. Fluorescence images were acquired using an Olympus IX-81 inverted microscope equipped with an Andor Revolution XD spinning disk system with FRAPPA, illumination and detection system Andor Revolution and Yokogawa CSU X1, EM-CCD Andor iXon X3 DU897 camera (software version Andor iQ3 v3.1), motorized x–y stage and a z-piezo stage, using a 20x objective (UPlansApo, NA 0.85, oil immersion). For visualization of MinD-Cy3 and MinE -Cy5 we used 561 nm and 640 nm laser lines and 617/73 band-pass and 690 long-pass filters respectively. Images were captured in multiple places at 30–60 s intervals. For the top-bottom correlation assay, images at the two planes were acquired at the same (x, y)-position. A shift of the Z-stage over the largest distance (57 μm) between observation planes was quicker than 0.1 s. Imaging of the membrane for quality control was done using 491 nm laser line and a 525/50 band-pass filter.

**Image sequence analysis**. Analysis of the image sequences was performed using a ImageJ (v1.52j) and custom Matlab (2016) scripts. Background correction and artefact removal was carried out as follows: First, for movies, frames were corrected for fluorescence bleaching by normalizing each frame on its mean intensity value. This corrected for the max. 20% intensity decay over long movies. Next, two correction images were processed: (1) A 'static background'-image `Imstat` was obtained by averaging out all moving (wave pattern) features of the movie stack and removing any residual background level. Thus, this image only contained static fluorescent features such as specks, holes and scratches. (2) An 'illumination correction' image `Imillum` was made by strongly smoothening out and averaging all movie frames and normalizing the result to its maximum. Finally, each movie

frames `Immovie` was corrected as via the following image operation: `Imcorrected = (Immovie − Imstat)/Imillum`. This way, irregularities are suppressed and wave amplitudes on the edge of each image are not underestimated compared to the amplitudes in the center of the image.

*Correlation histograms (Fig. 4E).* To quantify the synchronization of patterns patterns on the opposite membrane surfaces, we performed a correlation analysis of the fluorescence image time-lapse sequences. Because we found that spatial subregions within one field of view (FOV) exhibit different synchronization behavior (Fig. 4F), we divided each FOV into a grid of smaller subregions ($\approx 10 \times 10 \, \mu m^2$) and determined the temporal correlation between the top and bottom membrane individually for each subregion. The averaged correlation values from all subregions are collected in histograms shown in Fig. 4E. The data for the histograms is provided in the Source Data file.

The quantitative correlation analysis confirms the qualitative finding from visual inspection of the snapshots and kymographs in Fig. 4C. For low bulk height, the two membranes are almost perfectly synchronized in-phase (correlation close to 1). As the height increases, the distribution of correlation becomes bimodal as correlations close to −1 appear, indicating emerging regions of antiphase synchronization, which coexist with regions of in-phase synchronization. Antiphase synchronization reaches its maximum at 26 μm. For larger bulk heights, the correlation measure clusters around zero, indicating desynchronization of the patterns on the two opposite membranes.

*Length- and timescale analysis (Supplementary Figs. 10, 11).* Autocorrelation analysis to obtain the typical length- and timescales of the experimentally observed patterns (cf. Supplementary Fig. 9) was performed in Matlab. Spatial autocorrelation analysis was performed on 10 individual images per movie. For each autocorrelation output image, a radial average was recorded starting from the main central correlation peak. From the resulting spatial radial correlation curve we determined the first maximum indicating the dominant wavelength of the pattern, irrespective of propagation direction. For temporal correlation, we generated 20 $x$–$t$ or $y$–$t$ kymographs per movie (10 in $x$-direction and 10 in $y$-direction) evenly distributed over the center 0.7 fraction of an image. For each such kymograph, an autocorrelation analysis was performed. The $x = 0$ and $y = 0$ lines of these $x$–$t$ autocorrelation maps represent a temporal correlation curve averaged overall the original image points on the respective line. Next, these correlation curves were median averaged between different kymographs. Thus, the final correlation curve represents the average temporal correlation signal sampled from $20 \times 512$ surface locations. Analogous to the spatial correlation analysis, the first maximum after $t = 0$ indicates the dominant oscillation period. Experimental repeats of the same conditions (concentrations and height) were median averaged.

**Reporting summary**. Further information on research design is available in the Nature Research Reporting Summary linked to this article.

## Data availability
Data supporting the findings of this manuscript are available from the corresponding authors upon reasonable request. A reporting summary for this Article is available as a Supplementary Information file. Source data are provided with this paper.

## Code availability
Matlab codes used for experimental data analysis are provided in the repository https://github.com/jacobkers/MinED_patterns (doi:10.5281/zenodo.4637889). Numerical simulation codes (using COMSOL Multiphysics) and Mathematica codes for linear stability analysis and simulation data analysis are available at https://github.com/f-brauns/Min-bulk-surface-coupling/(doi:10.5281/zenodo.4644636).

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

## Acknowledgements

We thank Laeschkir Würthner for insightful discussions and support with numerical simulations, Federico Fanalista for help with microfabrication and inspiring discussions, and Yaron Caspi for providing purified Min proteins. E.F. acknowledges support by the German Excellence Initiative via the programme 'NanoSystems Initiative Munich' (NIM) and the Deutsche Forschungsgemeinschaft (DFG) via projects B02 within the Collaborative Research Center SFB1032 (Project-ID 201269156). F.B. acknowledges financial support by the DFG via the Research Training Group GRK2062 ('Molecular Principles of Synthetic Biology'). C.D. acknowledges support from the ERC Advanced Grant SynDiv (no. 669598) and the NanoFront and BaSyC programs.

## Author contributions

F.B., G.P., J.H., E.F., and C.D. designed research; G.P. and C.D. designed and carried out the experiments; F.B., J.H., and E.F. designed the theoretical models and performed the mathematical analyses; F.B., G.P., and J.K. analyzed data; and F.B., G.P., J.H., E.F., and C.D. wrote the paper.

## Funding

## Competing interests

The authors declare no competing interests.
