## [Peer Review File · Nature Communications]

Reviewer #1 (Remarks to the Author):

This is an excellent study, which demonstrates that the patterns in MinD, MinE system depend crucially on the bulk-surface ratio and intuitive explanations are provided for different observed regimes. The manuscript is very clearly presented and the experiments, simulations and analytical analyses are of high quality. I recommend publication after the following comments are addressed.

- 1.) Comment on what sets the characteristic bulk height H_C that marks the transition from the synchronized patterns without vertical oscillations to the membrane-to-bulk oscillation regime. Presumably, it is set by reaction rates and diffusion constants.
- 2.) Comment on how the presented results would be affected if the geometry was modified to the extended cylinders, which is closer to the in vivo geometry of the long mutant E.Coli, where cell division is disrupted. Would the membrane-to-membrane oscillations still be possible or would this regime disappear?
- 3.) Explain what the snapshots of experiments and 3D simulations are representing in Figs. 1, 3, S1, S9, and associated movies. Are they showing concentrations of MinD and MinE at a particular height (e.g. top, bottom, middle), or are they showing the concentrations integrated over the vertical height of the channel?
- 4.) Was the linear stability analysis in Figs. 2A and S3-S4 done in 1+2D or in 2+3D? The supplementary information describes the procedure for 1+2D, but it also states that the generalization to the 2+3D is straightforward.
- 5.) In captions of Figs. 4 and S5-S8 explain that the white color in overlays corresponds to the high concentrations on both top and bottom membrane.
- 6.) In Figure captions explain, which simulations were done in 2+3D and which in 1+2D.
- 7.) On page 2 in the SI explain the acronym SLB (supported lipid bilayer?).
- 8.) Authors should carefully check the references to figures.
 - * captions of Fig. 3: "(see Supplementary Fig. S8 for representative snapshots ...)" -> should be Fig. S9.
 - * page 5 in SI: "that connect low-density to high-density regions of the standing wave patterns (see Fig. S10)." -> should be S11
 - * page 11 in SI: "show the in-phase and anti-phase synchronization, or lack thereof, between the two opposite membranes (cf. Fig. 3)." -> should be Fig. 4
 - * captions of Fig. S9: "(cf. Movies S11–S14 and Fig. 5 in the main text)." -> should be Fig. 3
 - * captions of Movie S9: "(corresponding to the snapshots and kymographs in Fig. 3C)" -> should be Fig. 4D
- 9.) Typos:
 - * page 3: "The microchambers' height of directly controls ..."
 - * page 4: "For an intermediate bulk height (13 μm in Fig. 1C)," -> 13 μm should be 15 μm .
 - * page 8: "In a laterally extended system (bottom), exchange mass of mass ..."
 - * page 4 in SI: "... performed a correlation analysis of the of the fluorescence image time-lapse"

sequences."

Reviewer #2 (Remarks to the Author):

This paper studies pattern formation of the two proteins MinE and MinD from *Escherichia coli*. In vivo, these two proteins perform pole-to-pole oscillations, while on flat membranes in vitro they typically form traveling waves. It is commonly accepted that these dynamics, although they have different appearances, share the same underlying biochemical mechanisms where MinD dimerizes and binds to the membrane in the presence of ATP, while MinE binds to membrane-bound MinD to activate its ATPase activity. As a result, both proteins can detach from the membrane, with MinE still being able to linger on the membrane surface and activate another MinD dimer. While membrane-binding is not absolutely essential for dynamic instability and pattern formation it helps MinE to switch from a latent to an active form and thereby provides robustness to the protein patterns and helps to form regular protein waves. These individual steps are now well documented in a number of studies, from the structural to the in vivo level.

Furthermore, various in vitro studies have addressed the influence of salt concentration, membrane composition, flow, temperature, crowding, the presence of MinC, the role of MinE membrane binding and conformational switching (see summarized in Ramm et al 2019). In flow chambers, the Min system was found to organize into a variety of different dynamic patterns, such as amoebas, waves, swirls, mushrooms and bursts. The observed patterns were associated with distinct membrane protein densities, which was concluded to depend on the protein available in the bulk solution. Another recent study using untagged MinE found patterns that closely resemble "Turing patterns": spots, mesh, inverse spots, labyrinths and intermediate patterns. Furthermore, the assay demonstrated multistability in vitro, which was previously found in vivo (Wu et al 2016).

In addition, plenty of studies have looked at the role of confinement on Min protein pattern formation, either on 2, 2.5 or 3 dimensions (Schweizer et al, 2011, Zieske et al 2014, 2016), which have shown that confinement can lead to an oscillatory pattern in contrast to traveling waves on flat membranes. Together, these studies show that the Min system is now extremely well understood. However, it has been somewhat difficult to reconcile these different observations and, as the authors correctly write, there has been some disagreement as to which pattern most closely resembles the oscillation found in vivo.

In the current manuscript, the authors set out to "resolve the puzzling dichotomy". They use a novel kind of microchambers to study pattern formation of the Min system, where the bottom and top surfaces are covered with a supported membrane. These chambers could be prepared with defined heights to systematically increase the distance between these two membranes. The authors find that the Min proteins form different patterns at different inter-membrane distances: below a critical distance there is a close cross-talk between the pattern on the two membranes leading to a vertical synchronization of the patterns. This synchronization is lost above a certain threshold. In an intermediate regime there is an anti-correlation between the patterns before this correlation is lost above another threshold.

Interestingly, small inter-membrane distances lead to a protein pattern different to the more classical waves that can be found at large distances or filamentous *E. coli* cells. This pattern is somewhat reminiscent to the oscillatory behavior found in normal sized bacterial cells as well as to

the patterns found in membrane covered microwells, however, it does not require a lateral confinement. These experimental findings are in well agreement with an established theoretical model that is used here to understand the experimental observations.

According to the abstract and introduction, this paper aims to resolve the in vivo/in vitro conundrum of Min protein patterns. Unfortunately, the introduction is a great liability and weakness for the paper. In my view, the authors do not provide a convincing argument for why new experiments are required to understand Min pattern formation, furthermore, it needs to be better explained why and how previous in vitro reconstitution experiments failed to resemble in vivo observations. Finally, it sets the expectations for the paper very high, while the data and analysis have problems to deliver.

Below I am trying to explain several issues in more detail:

Several in vivo and in vitro studies made the strong case that the most important difference between the in vivo/in vitro patterns is the confinement of the proteins. According to the authors, however, three-dimensionally confined geometries cannot be used to distinguish in vitro and in vivo regimes, as confinement will result in a *resemblance* of an oscillation while in fact it is not. This is a very weak argument and authors should better explain their hypothesis, i.e. why would the in vivo oscillation not also be just the result of confinement as in the vitro case? why are the confined oscillations not in fact a close representation of the in vivo situation? This needs to be better explained.

This manuscript fails to provide strong arguments why the experimental setup used here is better suited than previous approaches. I.e. why are two membranes without lateral confinement a better representation of the E. coli cell than microfabricated chambers used by Zieske et al? The authors would need to better explain their reasoning as well as how exactly previous work failed to address the questions, because to this reader this is not obvious.

The author's main argument for a discrepancy of the in vivo/in vitro patterns is exemplified by the observation that Min proteins can show "homogeneous pulsing" under some conditions, a behavior that has not been found in vivo. However, this pulsing could also be explained by the perfect spherical symmetry of a lipid vesicle in vitro, which is most likely impossible to achieve for a living cell.

The authors claim they study the effect of bulk to surface ratio on Min protein pattern formation. I have a couple of reservations regarding this statement:

- Did they really study confinement or rather how the presence of a second membrane affects pattern formation?
- As the second membrane also acts as sink for the membrane binding proteins, it decreases the total number of proteins available to bind to the first membrane. As a result, the protein density is decreased given rise to a different pattern
- To study the effect of only confinement, the authors should in fact just limit the height of the buffer without having a second membrane there.

Apart from these serious problems that I have with the paper, the experimental data is actually interesting:

- the authors used a new experimental setting consisting of two flat membranes at a defined distance

- this allowed the users to study how the distance between the membrane influence protein patterns.
- below a critical distance there is perfect correlation between the pattern (there seems to be a discrepancy between the critical distances in Fig. 3D and Fig. 1C, can the authors clarify?)
- at intermediate distances there is anticorrelation
- above a critical distance any correlation is lost

This more sober summary would give an interesting paper, still relevant for our understanding of intracellular pattern formation. However, the paper would be relieved of the weight that the authors found a "new mechanism" of pattern formation. Accordingly, toning the abstract, introduction and conclusions down would significantly help the credibility of this paper.

Reviewer #3 (Remarks to the Author):

1. What are the major claims of the paper?

The authors demonstrate that the height (H_c) of the protein concentration (gradient) that vertically penetrates into the solution above the membrane surface can induce switches of the Min protein patterns *in vitro* and *in silico*. When the bulk height (H) is smaller than the penetration depth of the vertical concentration gradient (H_c), the Min protein patterns on both top and bottom surfaces will synchronize and move in phase. When H matches with H_c , the Min protein patterns on both top and bottom surfaces will synchronize, but oscillation on top and bottom surfaces is in anti-phase. When H is greater than H_c , the Min protein patterns on top and bottom surfaces will decouple and move out of phase. Since the Min protein oscillates from pole to pole across the cell length (ranging roughly from 2 to 5 μm) *in vivo*, the authors suggest that the Min protein patterns observed *in vitro* with a H value below 10 μm would mimic the confinement of a *E. coli* cell, i.e. the lateral movement represents synchronized in-phase oscillation and membrane-to membrane oscillation represents synchronized anti-phase oscillation (pole-to-pole oscillation) (Fig. 2A). Hence, the authors claim that the bulk-surface coupling reconciles the Min protein pattern formation *in vivo* and *in vitro* and addresses the underlying mechanism.

2. Are they novel and will they be of interest to others in the community and the wider field? If the conclusions are not original, it would be helpful if you could provide relevant references.

Experimental observation: The pattern formation of the Min system *in vitro* has been extensively studied. The complex mechanism underlying the pattern formation is also extensively addressed physicochemically and biochemically {Reviewed in Mizuuchi and Vecchiarelli (2018) *Phys. Biol.* 15, 031001; Ramm et. al. (2019) *Cell. Mol. Life Sci.* 76, 4245–4273}. The physical constraint, i.e. the bulk height, is discussed for the first time in this study.

Simulation: The authors state that the numerical model is based on the previous publication, Halatek and Frey, 2020 (*Nat. Physics* 14,741-752). The focus in this work is the effect of the bulk height.

Ideas:

- (1) The authors use the phase diagram to describe different Min protein patterns (Fig. 3).
- (2) Based on H vs H_c , the authors categorize the Min oscillation patterns into different regimes (Fig. 2).
- (3) Different Min protein patterns (Standing wave, Homogeneous oscillation, Traveling wave, and Amoeba) were reported for the first time in Ivanov and Mizuuchi, 2010 (PNAS 107: 8071–8). Vertical oscillation was implied in Vecchiarelli et al. 2016 (PNAS 113: E1479-E1488). The current work provides detailed qualitative features of the in vitro patterns and emphasizes on the bulk-surface coupling effect, which are advantageous.

3. Is the work convincing, and if not, what further evidence would be required to strengthen the conclusions?

- (1) While appreciating the qualitative views of the in vitro patterns, it is not straightforward to catch the connection to the pole-to-pole oscillation in vivo.
- (2) A brief discussion that includes the known factors to influence the Min pattern formation, such as time, lateral confinement, protein concentration ratio, lipid composition, and membrane surface defects, will be more comprehensive for our understanding of the contribution of the bulk height in the underlying mechanism. Although the bulk height (or the penetration depth of the protein concentration gradient) is a factor to influence the pattern formation, it is insufficient on its own to explain the mechanism underlying the Min protein pattern formation.
- (3) Two experiments may help to strengthen the work.
 - a. Images taken using the microchamber height(s) below $10\ \mu\text{m}$ in Fig. 4D,E,G, may draw better connection with the in vivo condition and better support the model in Fig. 2.
 - b. Images of the protein density taken inside the bulk volume as reported in Vecchiarelli et al., 2016 (PNAS 113: E1479-E1488) may support the coupling between vertical concentration gradient and bulk height.

4. On a more subjective note, do you feel that the paper will influence thinking in the field? The work reports the bulk height as a critical factor to sustain and switch between the Min oscillation patterns in vitro. Systematic investigation and discussion of the qualitative features of the Min protein patterns are useful especially for bridging with biologists.

5. Please feel free to raise any further questions and concerns about the paper.

Comments:

A. Terminology

1. (1) The meanings of the physical terms need to be clarified and make them consistent throughout, including the supplemental information. (2) Their correlations with different patterns (standing wave, homogeneous oscillation, traveling wave, amoeba) can be summarized in the same paragraph (or figure, or table). (3) For simplicity, if some terms refer to the same phenomenon, the same term is preferred. (4) It will also be helpful to specify which in vivo observations that they correlate with.

Stability-instability (chaos)-multistability

Lateral-vertical (local)

Symmetry-asymmetry

Commensurable-incommensurable

In-phase - anti-phase - (out-of-phase)

Phase chaos

Standing wave chaos

Defect-mediated turbulence

2. The correlations between terms under different regimes need to be clarified and consistent throughout, including the supplemental information.

$H_c > H$: No vertical oscillation; synchronized; in-phase; pole-to-pole oscillation; instability

$H_c \leq H$: Membrane-to-membrane-oscillation: coupled/synchronized; anti-phase; pole-to-pole oscillation; instability

$H_c \ll H$: Membrane-to bulk oscillation- decoupled oscillation; out-of-phase; instability

3. Definitions of H and H_c

H: bulk height vs chamber height (upper boundary?)

H_c : “penetration depth of vertical concentration gradient”, or “critical bulk height”?

4. P2, bottom: (1) The definition of the “bulk-surface ratio” as the “ratio of cytosolic bulk-volume to membrane surface” is confusing, since the phrase implies the geometrical ratio (volume/surface). Is it- the ratio of ‘the protein concentration’ in the ‘solution’ bulk volume and on the membrane surface? (2) If it refers to the protein concentration, which concentration- [MinD], [MinE], or [MinD-MinE]- is considered? (3) How does this correlate to the E:D ratio in all phase diagrams and in Fig. S9-11?

B. Title: The data to explain the bulk-surface coupling at the current state are not sufficient to support the correlation between the Min-protein pattern formation between in vitro and in vivo. Modification is suggested.

C. Other comments:

1. Abstract: “the Min protein dynamics on the membrane crucially depend on bulk gradients normal to the membrane”- (1) The authors need to clarify whether it is ‘bulk gradient’, or “bulk concentration”, or “bulk protein interaction”, or “bulk height” throughout the manuscript. (2) What is “normal to the membrane”?

2. P3, top: “The bulk-surface ratio is a measure for how far concentration gradients can penetrate into the cytosol.” (1) The depth of the concentration gradient penetrates into solution may be estimated experimentally by imaging across the bulk height (z-sectioning) that will reflect the transition state of the protein density in the bulk volume. This will be likely an addition to Fig. 4D,E,G. (2) ‘solution’ instead of cytosol. (3) How does the vertical concentration gradient form? Does the hypothesis imply that the self-organization also occurs in solution, thus giving rise to penetrating concentration gradient? (4) Just wonder, will it be difficult for the vertical gradient to form if the absolute protein concentration is too high or too low in solution?

3. P3, bottom: “lateral and local oscillations”- lateral and “vertical” oscillations?

4. P4, Results, paragraph#3, and P7, paragraph#4: It is confusing to have the terms ‘multistability’ and ‘instability’ refer to relating (or the same?) phenomena in two places.

5. P4, Results, paragraph#4, line 1: 15 μm in figure?

6. P4, Results, paragraph#4, lines 6-7: What is ‘a characteristic length scale’?

7. P5, Fig. 1, legend title: “pattern”

8. P6, paragraph#2, line 9: “underlying mesoscopic mechanisms (mass transport model)”. The Min proteins self-organize involving diffusion and various interactions, so personally don’t think it involves “transport”. Perhaps ‘migration’ or ‘movement’ may be easier to understand. Similar problem is identified in P9, paragraph#1, line 6.

9. P6, paragraph#4, lines 3-6: “For intermediate bulk heights (5–15 μm), we find nearly homogeneous oscillations, meaning large areas with a nearly homogeneous protein density that are phase separated by phase defect lines where the oscillator phase jumps.” (1) cite figure. (2) Physical

terms are used in description that is difficult to pick up what exactly 'phase separated waves' and 'phase defect lines' mean in the micrographs.

10. P7, paragraph#2: It will be more comprehensive to include other parameters in the system for discussion in order to bring up the significance of the bulk height and its contribution in the mechanism.

11. P7, "short" E:D ratio -> low

12. P8, Fig. 2: (1) The 'in vivo regime' in 2A does not cover $H < H_c$ and $H \geq H_c$ that both labeled with in vivo pole-to-pole in 2B. (2) "bulk gradient height" - I wonder if the protein molecules are considered in bulk, whether the gradient still exist? (3) clarify 'bulk height' or 'bulk concentration'. (4) Can author predict a value (or range) of H_c if possible? (5) Clarify whether 'no vertical' and/or 'membrane-to-membrane' oscillation reflect the pole-to-pole oscillation in vivo.

13. P9, paragraph#2: Here H_c is defined as 'critical bulk height', but earlier H_c appears to be the vertical penetration depth of the protein concentration gradient. Can authors clarify whether they are the same or different?

14. P11, Fig. 3: Can authors match the terms (standing waves, homogeneous oscillations, traveling waves, and amoeba) in Fig. 3 to (1) in vivo regime, transitional regime, and classical regime, and (2) conditions of $H < H_c$, $H \geq H_c$, and $H \gg H_c$ in Fig. 2? It will help us to switch terms and make connections between different purposes in different sections.

15. P11, Fig.3, legend, line 16: S9.

16. P12: "Interplanar pattern synchronization reveals vertical oscillation modes in experiments": "vertical oscillation modes underlie interplanar pattern synchronization"?

17. P12, paragraph#2, lines6: 'to' quantify.

18. P12 & P13, Fig. 4D-G: The following experiments may better support the claim to correlate with the in vivo condition: (1) Images taken using the bulk height(s) below 10 μm in Fig. 4D,E,G. (2) Images of the protein density taken inside the bulk volume as reported in Vecchiarelli et al., 2016.

19. P13, Fig. 4: (1) The label of the y axis in 4G: area coverage or overlapping area? (2) What does it imply to sum up the in-phase and anti-phase areas and present as 'total'? (3) legend of 4G: "Classification of top-bottom correlation as a function of bulk height,...". - Bulk height is not used in the correlation plot.

20. P14: It will be more comprehensive to include other parameters in the system for discussion in order to bring up the significance of the bulk height and its contribution in the mechanism.

21. Supplemental information:

a. Many editing errors.

b. Check units, figure numbers, matching statements in the text with correct figure numbers.

c. P2: in PDMS preparation section, the unit need to be corrected: mm instead of mM.

d. P3: The concentration unit needs to be checked and corrected. For example, the concentration of ADP (100 mM) in MinD protein preps seems very high. Also, 0.8 and 0.2 "M" MinD or MinE is not possible.

e. P6: Some parameters are not defined, such as cDT, cDD, ∂ .

f. P7: References in Table S1 may be helpful.

g. Fig. S5. Change 'left', 'center', 'right' to A, B, C.

h. Fig. S9. (1) line 3: Fig. 3 instead of Fig. 5 (2) Not sure E:D ratio – bulk-height – and patterns are matched to figure 1, 2, 3 etc.

i. Movie S17, the upper right video does not work.

j. Movie S19, how come cytosolic MinE is simulated instead of membrane-bound MinE?

k. Movie S20, the virtual cell width is at least 2-fold larger than normal E. coli. Please justify.

l. Line numbers will help the reviewers.

Yu-Ling Shih

Please find our point-by-point replies below. In the revised manuscript, passages with significant changes are highlighted in *red*.

Reviewer #1

This is an excellent study, which demonstrates that the patterns in MinD, MinE system depend crucially on the bulk-surface ratio and intuitive explanations are provided for different observed regimes. The manuscript is very clearly presented and the experiments, simulations and analytical analyses are of high quality. I recommend publication after the following comments are addressed.

We thank the reviewer for taking the time to carefully read our manuscript and for the very positive assessment of our work. We hope that our revisions in the manuscript and our point-by-point replies to the reviewer's comments answer all remaining questions.

1.) Comment on what sets the characteristic bulk height H_C that marks the transition from the synchronized patterns without vertical oscillations to the membrane-to-bulk oscillation regime. Presumably, it is set by reaction rates and diffusion constants.

The critical bulk height H_c depends on a subtle interplay of nucleotide exchange rate, diffusion constants, and reaction rates at the membrane. For a given set of these parameters, it is defined as the lowest bulk height where the membrane-to-membrane oscillation mode becomes unstable in the parameter plane of MinD and MinE concentrations. No simple mathematical expression for this is available.

In the revised manuscript, we changed the paragraph where we introduce H_c and explain its parameter dependence.

2.) Comment on how the presented results would be affected if the geometry was modified to the extended cylinders, which is closer to the in vivo geometry of the long mutant E.Coli, where cell division is disrupted. Would the membrane-to-membrane oscillations still be possible or would this regime disappear?

Extended cylinders, corresponding to filamentous E. coli cells, fall into the "low-bulk height" regime where only standing wave oscillations are found. Due to the cylinder geometry, the standing waves are confined to form along the cylinder axis. For sufficiently long cylinders, multiple wavelengths fit into the cell giving rise to "stripe oscillations" with multiple wave nodes (see Movie S19).

For membrane-to-membrane oscillations orthogonal to the cylinder axis, the "bulk-height" (essentially set by the cylinder diameter) would need to be significantly larger (about 5 μm), which clearly exceeds the values of E coli cells. In addition, the curved geometry might disfavor this oscillation mode versus oscillations along the cylinder axis [see Glock et al, eLife 2018]. Since this question of "curvature sensing" goes far beyond the scope of our work, which is

restricted to flat membrane surfaces, we decided not to comment on curved surfaces in our manuscript.

3.) Explain what the snapshots of experiments and 3D simulations are representing in Figs. 1, 3, S1, S9, and associated movies. Are they showing concentrations of MinD and MinE at a particular height (e.g. top, bottom, middle), or are they showing the concentrations integrated over the vertical height of the channel?

All figures show membrane concentrations, since the patterns form by proteins binding at the membranes. We have clarified this now in multiple pertinent passages in our manuscript.

4.) Was the linear stability analysis in Figs. 2A and S3-S4 done in 1+2D or in 2+3D? The supplementary information describes the procedure for 1+2D, but it also states that the generalization to the 2+3D is straightforward.

Because in 2+3D, we can simply pick a direction of interest in the x-y plane, the linear stability analysis is identical to the 1+2D case. We have clarified this point now in the SI.

5.) In captions of Figs. 4 and S5-S8 explain that the white color in overlays corresponds to the high concentrations on both top and bottom membrane.

We have revised the explanation of the color code in the caption of Fig. 4. We have added explanations in the respective captions in the SI.

6.) In Figure captions explain, which simulations were done in 2+3D and which in 1+2D.

We added this information to the figure captions as suggested.

7.) On page 2 in the SI explain the acronym SLB (supported lipid bilayer?).

Thanks for noting this. Indeed SLB is short for supported lipid bilayer. We added this information in the SI.

8.) Authors should carefully check the references to figures.

* captions of Fig. 3: "(see Supplementary Fig. S8 for representative snapshots ...)" -> should be Fig. S9.

* page 5 in SI: "that connect low-density to high-density regions of the standing wave patterns (see Fig. S10)." -> should be S11

* page 11 in SI: "show the in-phase and anti-phase synchronization, or lack thereof, between the two opposite membranes (cf. Fig. 3)." -> should be Fig. 4

* captions of Fig. S9: "(cf. Movies S11–S14 and Fig. 5 in the main text)." -> should be Fig. 3

* captions of Movie S9: "(corresponding to the snapshots and kymographs in Fig. 3C)" -> should be Fig. 4D

We greatly thank the referee for pointing us to these incorrect figure references. All references have been carefully checked and fixed in the revised manuscript.

9.) Typos:

* page 3: "The microchambers' height of directly controls ..."

* page 4: "For an intermediate bulk height (13 um in Fig. 1C)," -> 13 um should be 15 um.

* page 8: "In a laterally extended system (bottom), exchange mass of mass ..."

* page 4 in SI: "... performed a correlation analysis of the of the fluorescence image time-lapse sequences."

We thank the referee for carefully reading our manuscript and noting these typos, which we fixed in the revised manuscript.

Reviewer #2

We thank the reviewer for the feedback on our manuscript and the positive assessment of our experimental data. We hope that our revisions in the manuscript and our point-by-point replies below address the reviewer's concerns. In particular, we hope that we now convincingly communicate the point that our experimental findings, together with the theoretical analysis provide valuable novel insights into the dynamics underlying Min-protein pattern formation, and that these insights constitute significant progress towards resolving the *in vivo* vs *in vitro* dichotomy.

This paper studies pattern formation of the two proteins MinE and MinD from *Escherichia coli*. *In vivo*, these two proteins perform pole-to-pole oscillations, while on flat membranes *in vitro* they typically form traveling waves. It is commonly accepted that these dynamics, although they have different appearances, share the same underlying biochemical mechanisms where MinD dimerizes and binds to the membrane in the presence of ATP, while MinE binds to membrane-bound MinD to activate its ATPase activity. As a result, both proteins can detach from the membrane, with MinE still being able to linger on the membrane surface and activate another MinD dimer. While membrane-binding is not absolutely essential for dynamic instability and pattern formation it helps MinE to switch from a latent to an active form and thereby provides robustness to the protein patterns and helps to form regular protein waves. These individual steps are now well documented in a number of studies, from the structural to the *in vivo* level.

Furthermore, various *in vitro* studies have addressed at the influence of salt concentration, membrane composition, flow, temperature, crowding, the presence of MinC, the role of MinE membrane binding and conformational switching (see summarized in Ramm et al 2019). In flow chambers, the Min system was found to organize into a variety of different dynamic patterns, such as amoebas, waves, swirls, mushrooms and bursts. The observed patterns were associated with distinct membrane protein densities, which was concluded to depend on the protein available in the bulk solution. Another recent study using untagged MinE found patterns that closely resemble "Turing patterns": spots, mesh, inverse spots, labyrinths and intermediate patterns. Furthermore, the assay demonstrated multistability *in vitro*, which was previously found *in vivo* (Wu et al 2016).

In addition, plenty have studies have looked the role of confinement on Min protein pattern formation, either on 2, 2.5 or 3 dimensions (Schweizer et al, 2011, Zieske et al 2014, 2016), which have shown that confinement can lead to an oscillatory pattern in contrast to traveling waves on flat membranes. Together, these studies shows that the Min system is now extremely well well understood. However, it has been somewhat difficult to reconcile these different observations and, as the authors correctly write, there has been some disagreement as to which pattern most closely resembles to the oscillation found *in vivo*.

We agree with the referee that the Min system is *very well studied*, including several studies on the role of confinement. However, we respectfully disagree with the assertion that it is *well understood*.

A large variety of experimental studies has revealed a striking diversity of types of patterns exhibited by the Min system under various experimental conditions. As the reviewer notes, it is indeed difficult to reconcile these observations. A unifying understanding of the mechanisms and principles (beyond stating the biochemical reaction scheme) underlying the diverse pattern types and their relation among each other is still lacking. This becomes most apparent in the difficulty to reconcile the patterns found *in vitro* vs *in vivo*. In particular, the rich variety of patterns found *in vitro* suggests that there are a multitude of underlying oscillation modes.

A major point of our manuscript is that we identify these oscillation modes, explain the relations between them and conclude which is the one corresponding to the *in vivo* setting.

Regarding the role of physical confinement, all previous studies used geometries that inherently combined bulk confinement in the direction orthogonal to the membrane (“vertical confinement”) with lateral confinement. As we argue in the manuscript, and in our replies below, having vertical confinement of the bulk without lateral confinement turned out to be essential to disentangle the different modes underlying Min pattern formation.

In the current manuscript, the authors set out to “resolve the puzzling dichotomy”. They use a novel kind of microchambers to study pattern formation of the Min system, where the bottom and top surfaces are covered with a supported membrane. These chambers could be prepared with defined heights to systematically increase the distance between these two membranes. The authors find that the Min proteins form different patterns at different inter-membrane distances: below a critical distance there is a close cross-talk between the pattern on the two membranes leading to a vertical synchronization of the patterns. This synchronization is lost above a certain threshold. In an intermediate regime there is an anti-correlation between the patterns before this correlation is lost above another threshold.

Interestingly, small inter-membrane distances lead to a protein pattern different to the more classical waves that can be found at large distances or filamentous *E. coli* cells. This pattern is somewhat reminiscent to the oscillatory behavior found in normal sized bacterial cells as well as to the patterns found in membrane covered microwells, however, it does not require a lateral confinement. These experimental findings are in well agreement with an established theoretical model that is used here to understand the experimental observations.

According to the abstract and introduction, this paper aims to resolve the *in vivo/in vitro* conundrum of Min protein patterns. Unfortunately, the introduction is a great liability and weakness for the paper. In my view, the authors do not provide a convincing argument for why new experiments are required to understand Min pattern formation, furthermore, it needs to be better explained why and how previous *in vitro* reconstitution experiments failed to resemble in

vivo observations. Finally, it sets the expectations for the paper very high, while the data and analysis have problems to deliver.

Apparently, we failed to communicate clearly what our goals and claims are. In hindsight, we realized that indeed we perhaps focussed too much on the in vivo in vitro dichotomy in our original introduction and did not sufficiently explain some of the key concepts of our work. We therefore decided to completely rewrite our introduction from scratch. Correspondingly, we have also revised the discussion, to reflect the changes in the introduction. Moreover, we have changed the title to “Bulk-surface coupling identifies the mechanistic connection between Min-protein patterns in vivo and in vitro” and have revised the abstract.

Before addressing in detail the specific issues raised by the reviewer (see below), we would like to emphasize that our goal was not a one-to-one reconstruction of the in vivo system. As the reviewer notes, a qualitative (although not quantitative) resemblance on the phenomenological level has already been achieved in various in vitro experiments with 3D confining geometry. Rather, our goal was to gain insight into the *biophysical mechanisms* (mass-transport modes) governing Min-protein patterns and to provide a unification of the different regimes, which provides a mechanistic understanding instead of merely a phenomenological description. To that end, we designed our experimental setup to facilitate an investigation of Min pattern formation in a well-controlled setting that is amenable to a systematic theoretical analysis of those properties that previous theoretical studies identified to be most meaningful. Indeed, eliminating the confounding effects of lateral confinement and curved geometries was essential for this endeavour.

Below I am trying to explain several issues in more detail:

Several in vivo and in vitro studies made the strong case that the most important difference between the in vivo/in vitro patterns is the confinement of the proteins. According to the authors, however, three-dimensionally confined geometries cannot be used to distinguish in vitro and in vivo regimes, as confinement will result in a *resemblance* of an oscillation while in fact it is not. This is a very weak argument and authors should better explain their hypothesis, i.e. why would the in vivo oscillation not also be just the result of confinement as in the vitro case? Why are the confined oscillations not in fact a close representation of the in vivo situation? This needs to be better explained.

The referee is correct that, in principle, it would be possible that the in vivo oscillations are just a consequence of lateral confinement. However, we show that this is actually not the case.

Rather we show that the bulk-height confinement is the crucial factor, as it suppresses a fundamentally different oscillation mode that drives systems with larger bulk height. Disentangling these two distinct effects of confinement (lateral versus vertical) was only possible by eliminating the confounding effect of lateral confinement and investigating the role of vertical bulk-confinement by itself.

Whether confined oscillations in vitro as studied experimentally in [Zieske, Caspi, Wu] are a “representation” of the in vivo case depends on the bulk-surface ratio of the geometry. Because of the much larger wavelength in vitro, patterns in 3D confinements form only when the confinement dimensions are much larger than cells, both in the lateral dimensions and in the “vertical” dimension (i.e. orthogonal to the membrane). In this regime, the bulk-surface ratio of these confinements is large enough to potentially support vertical membrane-to-membrane oscillations and membrane-to-bulk oscillations. This is for example evidenced by the homogeneous pulsing in spherical vesicles [Litschel, 2018; Godino, 2019; Kohyama, 2019].

To address these points in the manuscript, we have substantially revised the introduction and discussion.

This manuscript fails to provide strong arguments why the experimental setup used here is better suited than previous approaches. I.e. why are two membranes without lateral confinement a better representation of the E. coli cell than microfabricated chambers used by Zieske et al? The authors would need to better explain their reasoning as well as how exactly previous work failed to address the questions, because to this reader this is not obvious.

As we already indicated above, our goal here is not to achieve a one-to-one representation of the in vivo system in vitro. Our setup allows a systematic control of the bulk height and the concentrations of MinD and MinE while eliminating confounding effects of lateral confinement. As we detail below in addressing the reviewer’s questions about our experimental setup, having two membrane surfaces parallel to each other allows us to unambiguously identify the different oscillation modes in the system owing to the signatures that they leave in the correlation between the patterns on the top and bottom membrane.

The author's main argument for a discrepancy of the in vivo/in vitro patterns is exemplified by the observation that Min proteins can show "homogeneous pulsing" under some conditions, a behavior that has not been found in vivo. However, this pulsing could also be explained by the perfect spherical symmetry of a lipid vesicle in vitro, which is most likely impossible to achieve for a living cell.

Note that there are spherical “minicells” (which are actually the Min-proteins’ namesake), but these cells have never been observed to show “homogeneous pulsing” [see Corbin 2002, Shih 2005].

Furthermore, we would like to note that the “perfect spherical symmetry” is neither the cause nor a requirement for blinking as observed in vitro in GUVs. The blinking is a consequence of the spatially uniform mode becoming oscillatory unstable, which requires a sufficiently large diameter (equivalent to bulk height in case of the flat in vitro geometry). Spatial perturbations of the spherical geometry will result in spatial inhomogeneities of the chemical concentration on the membrane. Even if the perturbation would be ellipsoidal and large enough to destabilize a *non-uniform* oscillatory mode, there would be no guarantee that this mode would suppress the uniform oscillation in favor of a spatial oscillation (e.g. pole-to-pole). In contrast, for minicells no

mode is active/unstable and therefore no patterns are observed. Only if the cell size (length) exceeds a certain threshold, the mode corresponding to pole-to-pole oscillations becomes unstable (see Halatek & Frey, Cell Reports, 2012 for details).

The authors claim they study the effect of bulk to surface ratio on Min protein pattern formation. I have a couple of reservations regarding this statement:

- Did they really study confinement or rather how the presence of a second membrane affects pattern formation?

We did both, and that's one of the key points of the paper, as it demonstrates the role of vertical concentration gradients in the bulk which couples both membranes.

In the low bulk height regime, both membranes are tightly coupled and the patterns that form are always highly correlated between both membranes. Thus, the system is top-bottom symmetric. This top-bottom symmetry at low bulk height is analogous to the rotational symmetry of pole-to-pole oscillations in cylindrical cells. Moreover, the top-bottom symmetry guarantees that there is no flux across a plane at half bulk height, parallel to the membranes. Thus, the system is analogous to a system with a single membrane and a no-flux (i.e. reflective) boundary placed at half bulk height. Vice versa, in the large bulk height regime, the membranes are fully decoupled and therefore the presence of the second membrane does not matter. Only for intermediate bulk heights, the two membranes interact in a non-trivial way, giving rise to membrane-to-membrane oscillations.

- As the second membrane also acts as sink for the membrane binding proteins, it decreases the total number of proteins available to bind to the first membrane. As a result, the protein density is decreased given rise to a different pattern

This is accounted for in the analysis. The protein density per membrane surface is equivalent to that of a setup with one membrane and an inert surface at half the bulk height.

- To study the effect of only confinement, the authors should in fact just limit the height of the buffer without having a second membrane there.

Indeed, such geometry might be interesting as an additional control. Unfortunately, Min proteins stick to almost all surfaces (including all materials suitable for the construction of microchambers). Achieving a fully inert non-sticking surface is therefore impossible in practice. Therefore, two membranes are the best way to confine the volume in a controlled manner.

Luckily, having two parallel membrane surfaces turns out to be very useful since the correlations of the patterns on the two surfaces serve as hallmarks of the different oscillation modes. The onset of vertical membrane-to-membrane oscillations clearly marks the onset of a qualitatively new regime.

Apart from these serious problems that I have with the paper, the experimental data is actually interesting:

- the authors used a new experimental setting consisting of two flat membranes at a defined distance
- this allowed the users to study how the distance between the membrane influence protein patterns.
- below a critical distance there is perfect correlation between the pattern (there seems to be a discrepancy between the critical distances in Fig. 3D and Fig. 1C, can the authors clarify?)
- at intermediate distances there is anticorrelation
- above a critical distance any correlation is lost

We thank the reviewer for acknowledging the interest of our experimental findings, as well as summarizing a list of interesting observations. Let us reiterate that these experimental findings enabled us, in combination with the modeling and theory, to gain important new insights into the role of bulk concentration gradients for Min-protein pattern formation.

Regarding the question about the critical “distance”. We assume that the reviewer means the critical bulk height here. Note that the critical bulk height indicated in Fig. 3D takes into account multistability, specifically the fact that both standing waves and hom. oscillations are found for bulk heights of 6 and 8 μm for an E:D ratio of 1:1. The regime of multistability is also indicated in Fig. 1C,D by the gray background shading.

This more sober summary would give an interesting paper, still relevant for our understanding of intracellular pattern formation. However, the paper would be relieved of the weight that the authors found a "new mechanism" of pattern formation. Accordingly, toning the abstract, introduction and conclusions down would significantly help the credibility of this paper.

For clarification: It was not our intent to claim that we found a “new mechanism” of pattern formation in general. Rather, we identify the mechanisms underlying pattern formation in the Min system in different regimes, as a function of the bulk-surface ratio. In particular, we explain the role of vertical bulk concentration gradients in these mechanisms. Thus, our work highlights the importance of bulk gradients for pattern formation in the Min system (and in other systems with bulk-surface coupling in general). This has important implications for the in vivo vs in vitro conundrum presented by the Min system, as the cellular confinement significantly restricts bulk gradients, which suppresses vertical oscillation modes. We therefore conclude that pattern formation in vivo operates by a fundamentally different mechanism compared to the typical in vitro settings, where the much larger bulk volume allows for the formation of significant bulk concentration gradients giving rise to vertical oscillation modes.

We have completely rewritten the introduction from scratch, we have rewritten the title and abstract, as well as the discussion to clarify what our findings and claims are.

Reviewer #3 (Remarks to the Author)

First of all, we would like to thank the reviewer for her careful reading of our manuscript and SI and for her very detailed feedback. We appreciate the great effort that must have gone into writing this detailed report.

1. What are the major claims of the paper?

The authors demonstrate that the height (H_c) of the protein concentration (gradient) that vertically penetrates into the solution above the membrane surface can induce switches of the Min protein patterns in vitro and in silico. When the bulk height (H) is smaller than the penetration depth of the vertical concentration gradient (H_c), the Min protein patterns on both top and bottom surfaces will synchronize and move in phase. When H matches with H_c , the Min protein patterns on both top and bottom surfaces will synchronize, but oscillation on top and bottom surfaces is in anti-phase. When H is greater than H_c , the Min protein patterns on top and bottom surfaces will decouple and move out of phase. Since the Min protein oscillates from pole to pole across the cell length (ranging roughly from 2 to 5 μm) in vivo, the authors suggest that the Min protein patterns observed in vitro with a H value below 10 μm would mimic the confinement of a *E. coli* cell, i.e. the lateral movement represents synchronized in-phase oscillation and membrane-to membrane oscillation represents synchronized anti-phase oscillation (pole-to-pole oscillation) (Fig. 2A). Hence, the authors claim that the bulk-surface coupling reconciles the Min protein pattern formation in vivo and in vitro and addresses the underlying mechanism.

We appreciate the reviewer's succinct summary of our results. A particular point the reviewer makes and that we would like to highlight is that it is the confinement of the bulk volume that restricts the formation of vertical gradients (equivalently, radial gradients in cylinder geometry)

2. Are they novel and will they be of interest to others in the community and the wider field? If the conclusions are not original, it would be helpful if you could provide relevant references.

Experimental observation: The pattern formation of the Min system in vitro has been extensively studied. The complex mechanism underlying the pattern formation is also extensively addressed physicochemically and biochemically {Reviewed in Mizuuchi and Vecchiarelli (2018) *Phys. Biol.* 15, 031001; Ramm et. al. (2019) *Cell. Mol. Life Sci.* 76, 4245–4273}. The physical constraint, i.e. the bulk height, is discussed for the first time in this study.

We thank the referee for emphasizing that our study is the first to address the role of the bulk confinement using a combination of experiments and theoretical analysis. Moreover, we would like to stress that while the Min system has indeed been extensively studied in experiments, a unifying mechanistic understanding of the underlying pattern-forming mechanisms is still lacking. The goal work is to provide such an understanding.

Simulation: The authors state that the numerical model is based on the previous publication, Halatek and Frey, 2020 (Nat. Physics 14,741-752). The focus in this work is the effect of the bulk height.

Ideas:

- (1) The authors use the phase diagram to describe different Min protein patterns (Fig. 3).
- (2) Based on H vs H_c , the authors categorize the Min oscillation patterns into different regimes (Fig. 2).
- (3) Different Min protein patterns (Standing wave, Homogeneous oscillation, Traveling wave, and Amoeba) were reported for the first time in Ivanov and Mizuuchi, 2010 (PNAS 107: 8071–8). Vertical oscillation was implied in Vecchiarelli et al. 2016 (PNAS 113: E1479-E1488). The current work provides detailed qualitative features of the in vitro patterns and emphasizes on the bulk-surface coupling effect, which are advantageous.

We thank the reviewer for this summary. We would like to point out that transient homogeneous oscillations, indicating the presence of the vertical membrane-to-bulk oscillation mode, can already be seen in Movie S1 in [Ivanov and Mizuuchi, 2010]. These oscillations were only transient, however, and they transitioned to traveling waves after a few oscillation cycles.

Vertical oscillations were studied theoretically [Halatek and Frey, 2018] and observed in vesicles [Litschel, 2018; Godino, 2019; Kohyama, 2019]. These oscillations are driven by a membrane-to-bulk transport mode, which is fundamentally different from the lateral mass-transport mode that drives pole-to-pole oscillations in vivo. Importantly, we reveal a second vertical oscillation mode that is specific to the two-membrane setup: membrane-to-membrane oscillations.

Finally, we would like to note that while previous work has suggested that in vitro Min patterns may be related to vertical oscillations, simply due to the fact that proteins shuffle back and forth between membrane and the bulk, we are here interested in a precise mathematical characterisation of the role bulk gradients play for pattern formation, which we provide here for the first time.

3. Is the work convincing, and if not, what further evidence would be required to strengthen the conclusions?

- (1) While appreciating the qualitative views of the in vitro patterns, it is not straightforward to catch the connection to the pole-to-pole oscillation in vivo.
- (2) A brief discussion that includes the known factors to influence the Min pattern formation, such as time, lateral confinement, protein concentration ratio, lipid composition, and membrane surface defects, will be more comprehensive for our understanding of the contribution of the bulk height in the underlying mechanism. Although the bulk height (or the penetration depth of the protein concentration gradient) is a factor to influence the pattern formation, it is insufficient on its own to explain the mechanism underlying the Min protein pattern formation.

Ad (1): Indeed, the connection to the pole-to-pole oscillation is on a conceptual, rather than a phenomenological level. We hope that our substantial revisions in the manuscript help to clarify the connection.

Ad (2): Of course, Min pattern formation is affected by many factors, and we did not intend to claim that the bulk height is the only factor that influences these patterns. Our goal is not to explain Min pattern formation from the ground up, but to investigate and explain the specific role of bulk-surface coupling (and hence the bulk height). The key message of our manuscript is that a seemingly insignificant *physical constraint* (the height of the bulk solution) has such a dramatic influence on protein-pattern formation, which has so far been predominantly studied through the lens of biochemical interactions. Keeping all other parameters constant, varying only the bulk height gives rise to a range of fundamentally different pattern types (as evidenced by their synchronization behavior between the two membrane surfaces).

(3) Two experiments may help to strengthen the work.

a. Images taken using the microchamber height(s) below 10 μm in Fig. 4D,E,G, may draw better connection with the *in vivo* condition and better support the model in Fig. 2.

In the experiments probing top-bottom correlation of patterns, we found the transition from in-phase synchronization to anti-phase synchronization at a height of about 20 μm . For all lower bulk heights, we find a near-complete in-phase synchronization (see 10 μm in Fig. 4, as well as data for 5 μm in the figure below). Because there is no qualitative change in the behavior below 10 μm , we decided to not include these data in the manuscript.

In-phase synchronization of protein patterns for 5 μm bulk height. From left to right: Fluorescence images take at the top-membrane, bottom-membrane, overlay.

The observation of in-phase synchronization below a critical bulk height is exactly what we expect based on our theory and simulations. A quantitative comparison of the critical bulk height to the phase diagrams in Fig. 2 and Fig. 3 is confounded by multistability, i.e. the occurrence of distinct pattern types for the same experimental conditions / simulation parameters.

b. Images of the protein density taken inside the bulk volume as reported in Vecchiarelli et al., 2016 (PNAS 113: E1479-E1488) may support the coupling between vertical concentration gradient and bulk height.

Indeed such data would be great to have. Unfortunately, there are several technical difficulties that push acquiring such data way beyond the scope of the current manuscript. The z-resolution of confocal microscopes is much worse than the x-y resolution. Close to the membrane, this makes it hard to distinguish the signal from proteins in solution from that coming from membrane-bound proteins. Since the typical penetration depth of bulk gradients is on the order of a few microns, they cannot be resolved with our setup.

4. On a more subjective note, do you feel that the paper will influence thinking in the field?

The work reports the bulk height as a critical factor to sustain and switch between the Min oscillation patterns in vitro. Systematic investigation and discussion of the qualitative features of the Min protein patterns are useful especially for bridging with biologists.

We thank the reviewer for emphasizing the importance of our results. We are particularly happy that the reviewer lauds our efforts to bridge with biologists. A key goal of our paper is to communicate the relevance of physical parameters and constraints, such as the bulk height, for protein-based pattern formation.

5. Please feel free to raise any further questions and concerns about the paper.

Comments:

A. Terminology

1. (1) The meanings of the physical terms need to be clarified and make them consistent throughout, including the supplemental information. (2) Their correlations with different patterns (standing wave, homogeneous oscillation, traveling wave, amoeba) can be summarized in the same paragraph (or figure, or table). (3) For simplicity, if some terms refer to the same phenomenon, the same term is preferred. (4) It will also be helpful to specify which in vivo observations that they correlate with.

Stability-instability (chaos)-multistability

Lateral-vertical (local)

Symmetry-asymmetry

Commensurable-incommensurable

In-phase - anti-phase - (out-of-phase)

Phase chaos

Standing wave chaos

Defect-mediated turbulence

Throughout the revised manuscript, we have made an effort to clarify technical terms and avoid jargon. Following the comment of the reviewer, we now decided to forgo some terms, such as

commensurability, which are not essential in the scope of the present manuscript. Note, furthermore, that some terms, like phase chaos and defect-mediated turbulence, are common terms in the pattern-formation literature, where they describe specific phenomena (see, e.g. Refs. 23, 24). Using these terms is important to connect our work to this community.

2. The correlations between terms under different regimes need to be clarified and consistent throughout, including the supplemental information.

$H_c > H$: No vertical oscillation; synchronized; in-phase; pole-to-pole oscillation; instability

$H_c \leq H$: Membrane-to-membrane-oscillation: coupled/synchronized; anti-phase; pole-to-pole oscillation; instability

$H_c \ll H$: Membrane-to bulk oscillation- decoupled oscillation; out-of-phase; instability

We hope that our revisions throughout the manuscript and SI (also in response to the reviewers comments below and the comments by reviewers 1 and 2) have helped to clarify the meanings of these terms and the relations between them. If something specific remains unclear, we would of course be glad to hear that.

3. Definitions of H and H_c

H: bulk height vs chamber height (upper boundary?)

Bulk height and (micro)chamber height refer to the same thing. We use microchamber height when we explicitly talk about the experimental system.

H_c : “penetration depth of vertical concentration gradient”, or “critical bulk height”?

H_c is the critical bulk height, defined as the lowest bulk height at which the vertical membrane-to-membrane oscillation mode becomes unstable. This critical bulk height is (approximately) determined by the penetration depth of vertical concentration gradients.

4. P2, bottom: (1) The definition of the “bulk-surface ratio” as the “ratio of cytosolic bulk-volume to membrane surface” is confusing, since the phrase implies the geometrical ratio (volume/surface). Is it- the ratio of ‘the protein concentration’ in the ‘solution’ bulk volume and on the membrane surface? (2) If it refers to the protein concentration, which concentration- [MinD], [MinE], or [MinD-MinE]- is considered? (3) How does this correlate to the E:D ratio in all phase diagrams and in Fig. S9-11?

By bulk-surface ratio we do in fact mean the geometric ratio of cytosolic bulk volume to membrane surface. It is *not* the ratio of protein concentrations and therefore does not correlate with the E:D ratio which is set by the total concentrations of MinE and MinD in the system.

B. Title: The data to explain the bulk-surface coupling at the current state are not sufficient to support the correlation between the Min-protein pattern formation between in vitro and in vivo. Modification is suggested.

We thank the referee for suggesting us to choose a different title. We have revised the title to “Bulk-surface coupling identifies the mechanistic connection between Min protein patterns in vivo and in vitro”.

C. Other comments:

1. Abstract: “the Min protein dynamics on the membrane crucially depend on bulk gradients normal to the membrane”- (1) The authors need to clarify whether it is ‘bulk gradient’, or “bulk concentration”, or “bulk protein interaction”, or “bulk height” throughout the manuscript. (2) What is “normal to the membrane”?

Ad (1): Bulk gradient is short for bulk concentration gradient; bulk height is short for the height of the microchamber. We have revised the abstract to avoid potential confusion.

Ad (2): In the context of geometry, normal means orthogonal to the membrane plane (in our setup = vertical). To avoid potential confusion, we now use the word orthogonal throughout the manuscript.

2. P3, top: “The bulk-surface ratio is a measure for how far concentration gradients can penetrate into the cytosol.” (1) The depth of the concentration gradient penetrates into solution may be estimated experimentally by imaging across the bulk height (z-sectioning) that will reflect the transition state of the protein density in the bulk volume. This will be likely an addition to Fig. 4D,E,G. (2) ‘solution’ instead of cytosol. (3) How does the vertical concentration gradient form? Does the hypothesis imply that the self-organization also occurs in solution, thus giving rise to penetrating concentration gradient? (4) Just wonder, will it be difficult for the vertical gradient to form if the absolute protein concentration is too high or too low in solution?

Ad (1): Indeed such data would be nice to have. Unfortunately, there are several technical difficulties as we explain in our reply to comment 3.(3).

Ad (2): We have changed the term “cytosol” to “bulk solution”.

Ad (3): Patterns form on the membrane by attachment and detachment of proteins. Attachment and detachment processes necessarily lead to bulk gradients, as detachment increases the bulk concentration in the membrane’s vicinity, and attachment depletes the cytosol there. We have revised the introduction to clarify this important point.

Ad (4): Vertical gradients of MinD concentrations are always present due to nucleotide exchange. As MinD predominantly attaches in its ATP-bound form and detaches in its ADP-bound form, the membrane acts as a sink for MinD-ATP and a source for MinD-ADP. This is the case even in steady state, since the individual proteins continually cycle between cytosol and membrane (in an ATP-dependent manner). In addition to this stationary gradient, there are temporally changing gradients when the system is in a dynamic state (oscillations or wave patterns). Such dynamics are only observed for sufficiently high protein concentrations; see for instance Fig. S4.

3. P3, bottom: “lateral and local oscillations”- lateral and “vertical” oscillations?

We suppose that you suggest to include the word “vertical” here to account for the fact that vertical membrane-to-membrane oscillations in our microchambers are in some sense analogous to pole-to-pole oscillations. However, at this point in the manuscript, we would like to emphasize the key difference between the low bulk height regime (which corresponds to the *in vivo* case both in terms of the bulk-surface ratio and the pattern formation modes) and the large bulk height regime, corresponding to traditional the *in vitro* setting.

4. P4, Results, paragraph#3, and P7, paragraph#4: It is confusing to have the terms ‘multistability’ and ‘instability’ refer to relating (or the same?) phenomena in two places.

The terms instability and multistability refer to different phenomena. Instability refers to the tendency of small perturbations of a (homogeneous) steady state to grow, thus leading to pattern formation. Multistability means that for a single combination of parameters, different states of the system are stable, i.e. persist even after (small) perturbations.

5. P4, Results, paragraph#4, line 1: 15 μm in figure?

We have fixed this typo.

6. P4, Results, paragraph#4, lines 6-7: What is ‘a characteristic length scale’?

Here, by “characteristic length scale” we mean that the pattern has a wavelength that has the same value at different spatial positions and is maintained over many oscillation periods. To clarify, we have replaced the words “length scale” with “wavelength” in the revised manuscript.

7. P5, Fig. 1, legend title: “pattern”

We have fixed this typo.

8. P6, paragraph#2, line 9: “underlying mesoscopic mechanisms (mass transport model)”. The Min proteins self-organize involving diffusion and various interactions, so personally don’t think it involves “transport”. Perhaps ‘migration’ or ‘movement’ may be easier to understand. Similar problem is identified in P9, paragraph#1, line 6.

In the presence of concentration gradients, diffusion leads to a transport of mass. Concentration gradients are caused by the reaction kinetics at the membrane, which, in concert with diffusion, leads to self-organized directed transport. The theory we developed [ref Nat Phys, PRX] explains pattern formation as such a self-organized transport process, where mass-transport modes play a central role. We believe that this is the correct physical term to use here.

9. P6, paragraph#4, lines 3-6: “For intermediate bulk heights (5–15 μm), we find nearly homogeneous oscillations, meaning large areas with a nearly homogeneous protein density that are phase separated by phase defect lines where the oscillator phase jumps.” (1) cite figure. (2) Physical terms are used in description that is difficult to pick up what exactly ‘phase separated waves’ and ‘phase defect lines’ mean in the micrographs.

We have added references to Figure 1D throughout the paragraph. We have also removed the potentially confusing passage “that are phase separated by phase defect lines where the oscillator phase jumps”.

10. P7, paragraph#2: It will be more comprehensive to include other parameters in the system for discussion in order to bring up the significance of the bulk height and its contribution in the mechanism.

We have completely revised this paragraph. Rather than a comprehensive discussion, the aim of this paragraph is to redirect the readers focus to the fundamental pattern-forming mechanisms. We therefore believe that this is not the right place for a discussion of other parameters.

11. P7, “short” E:D ratio -> low

Indeed, the phrase “short E:D ratio”, which is meant to define the abbreviation “E:D ratio” was ambiguous. We have rephrased to “*E:D ratio* for brevity” to avoid confusion.

12. P8, Fig. 2: (1) The ‘in vivo regime’ in 2A does not cover $H < H_c$ and $H \geq H_c$ that both labeled with in vivo pole-to-pole in 2B. (2) “bulk gradient height”- I wonder if the protein molecules are considered in bulk, whether the gradient still exist? (3) clarify ‘bulk height’ or ‘bulk concentration’. (4) Can author predict a value (or range) of H_c if possible? (5) Clarify whether ‘no vertical’ and/or ‘membrane-to-membrane’ oscillation reflect the pole-to-pole oscillation in vivo.

Ad (1): We are not entirely certain that we understand the reviewer’s comment correctly. The in vivo regime in Fig. 2A spans from $H=0$ to $H=H_c$ and is characterised by the *absence* of vertical oscillations across the bulk. Fig. 2B serves as a visual aid to connect the lateral pole-to-pole transport in vivo with the lateral transport in the in vitro system studied here for $H < H_c$.

Ad (2): We ran a search but did not find any instances of “bulk gradient height” in our manuscript. Maybe the reviewer is referring to the term penetration depth? In any case, our model considers molecules to be in bulk once they undergo the detachment reaction step. The bulk gradient is a consequence of this reaction step, not a prerequisite.

Ad (3): Bulk height refers to the distance between the both opposite membranes; see the label “bulk height” in Fig. 1A and ‘H’ in Fig. 2B. Bulk concentration refers to the chemical concentration of bulk proteins.

Ad (4): From our model, we find a value of approximately $5 \mu\text{m}$ for H_c , as can be read of from Fig. 2A and is stated in the paragraph “Intermediate bulk height”. However, the precise value depends on other parameters (reaction rates, bulk diffusion constants).

Ad (5): Both the lateral oscillation and the vertical membrane-to-membrane oscillation correspond to the pole-to-pole oscillation in vivo, see Fig. 2BC. In the latter case, a laterally isolated column of bulk solution with the two membrane patches at its top and bottom corresponds to the cell with its two poles. Since in the experimental system, the two membrane surfaces are laterally extended they constitute a continuum of coupled oscillators. In contrast, the cell is only a single oscillator. We have clarified this important point in the manuscript.

13. P9, paragraph#2: Here H_c is defined as ‘critical bulk height’, but earlier H_c appears to be the vertical penetration depth of the protein concentration gradient. Can authors clarify whether they are the same or different?

H_c is indeed the critical bulk height. It is defined as the lowest bulk height where vertical membrane-to-membrane oscillations are found for a specific set of reaction rates and diffusion constants (but allowing adjustment of total protein concentrations). We have clarified this in the revised manuscript.

Heuristically, the vertical penetration depth is what sets this critical height. However, due to the nonlinearities in the attachment-detachment terms at the membrane, which cause the vertical gradients, there is no unique way of defining a vertical penetration depth. Thus, this heuristic concept is hard to quantify precisely.

14. P11, Fig. 3: Can authors match the terms (standing waves, homogeneous oscillations, traveling waves, and amoeba) in Fig. 3 to (1) in vivo regime, transitional regime, and classical regime, and (2) conditions of $H < H_c$, $H \geq H_c$, and $H \gg H_c$ in Fig. 2? It will help us to switch terms and make connections between different purposes in different sections.

The conditions $H < H_c$, $H \geq H_c$, and $H \gg H_c$ denote where specific modes become unstable, i.e. where they start to contribute to pattern formation. However, it is important to note that there is a large part of the parameter space, where not only one but multiple modes are operational. As a consequence of this “overlap” there is multistability, i.e. different patterns can form for the same set of conditions. We have tried to make this clear using colors to reflect the different patterns in the phase diagram.

The terms (i) “in vivo regime”, (ii) “transitional regime” and (iii) “classical regime”, respectively, denote the parameter regions where (i) only the lateral oscillation mode is operational, (ii) the vertical membrane-to-membrane mode (and largely also the lateral mode) are operational, and (iii) where the vertical membrane-to-bulk mode and the lateral mode are operational.

We have revised the manuscript to emphasize the aspect of multistability in the relationship between the different oscillation mechanisms, their regimes of operation and the observed pattern types.

Finally, the amoeba patterns are found in experiments only in a regime where the mathematical model that we employ is not valid. We emphasize this both in the main text and in the caption of Fig. 3, and comment on potential model extensions that might also cover the “amoeba” regime.

15. P11, Fig.3, legend, line 16: S9.

We have fixed this typo.

16. P12: “Interplanar pattern synchronization reveals vertical oscillation modes in experiments”: “vertical oscillation modes underlie interplanar pattern synchronization”?

We have chosen the phrasing here to emphasize that the observed synchronization is evidence for the vertical oscillation modes and thus for the vertical bulk gradients.

17. P12, paragraph#2, lines6: ‘to’ quantify.

We have fixed this typo.

18. P12 & P13, Fig. 4D-G: The following experiments may better support the claim to correlate with the in vivo condition: (1) Images taken using the bulk height(s) below 10 μm in Fig. 4D,E,G. (2) Images of the protein density taken inside the bulk volume as reported in Vecchiarelli et al., 2016.

See above for our detailed comments on these suggestions.

19. P13, Fig. 4: (1) The label of the y axis in 4G: area coverage or overlapping area? (2) What does it imply to sum up the in-phase and anti-phase areas and present as ‘total’? (3) legend of 4G: “Classification of top-bottom correlation as a function of bulk height,... “.- Bulk height is not used in the correlation plot.

Ad (1): Area coverage of regions with the respective correlation type (in-phase, anti-phase).

Ad (2): The sum indicated the area coverage of regions that are either in-phase or anti-phase correlated. Since this can obviously lead to confusion, we have decided to omit the term “total correlation”.

Ad (3): The bulk height is plotted on the x-axis, as indicated in Fig. 4G.

20. P14: It will be more comprehensive to include other parameters in the system for discussion in order to bring up the significance of the bulk height and its contribution in the mechanism.

While the paper is already at maximum length limit for the journal, we agree that an investigation of other parameters would be interesting. However, individual kinetic rates can rarely be changed experimentally without affecting other processes. We therefore focused on those parameters that can be controlled well experimentally, and have a clear effect on the dynamics that can be understood within our theory. Those are the microchamber height (= bulk height) and the protein concentrations. All other parameters that affect the patterns are kept constant. This emphasizes that changing the microchamber height alone can drive transitions between fundamentally different phenomena, and therefore is an essential parameter.

21. Supplemental information:

- a. Many editing errors.
- b. Check units, figure numbers, matching statements in the text with correct figure numbers.
- c. P2: in PDMS preparation section, the unit need to be corrected: mm instead of mM.
- d. P3: The concentration unit needs to be checked and corrected. For example, the concentration of ADP (100 mM) in MinD protein preps seems very high. Also, 0.8 and 0.2 “M” MinD or MinE is not possible.
- e. P6: Some parameters are not defined, such as cDT, cDD, ∂ .
- f. P7: References in Table S1 may be helpful.
- g. Fig. S5. Change ‘left’, ‘center’, ‘right’ to A, B, C.
- h. Fig. S9. (1) line 3: Fig. 3 instead of Fig. 5 (2) Not sure E:D ratio – bulk-height – and patterns are matched to figure 1, 2, 3 etc.

We thank the referee for carefully reading our manuscript and for suggesting these improvements and pointing us to various typos. We have carefully read the SI again to fix editing errors and typos. In addition, we have revised some passages in the SI according to the above suggestions where appropriate.

- i. Movie S17, the upper right video does not work.

Indeed, the stationary top-right panel in the video is not broken. The phase portrait of a traveling-wave pattern appears stationary, since each point in space traverses the same oscillation cycle as the wave propagates through the system. Put differently, in a coordinate frame moving with the same speed as the wave, the wave appears stationary. We have added a clarifying remark in the movie description.

- j. Movie S19, how come cytosolic MinE is simulated instead of membrane-bound MinE?

In all our simulations, all components are simulated. In movie S19, we show one representative membrane concentration (MinD + MinD-MinE) and one representative cytosol concentration (MinE).

- k. Movie S20, the virtual cell width is at least 2-fold larger than normal E. coli. Please justify.

The point of Movie S20 is to show that pole-to-pole oscillations persist at a cell radius that corresponds to the lowest bulk-surface ratio realized in our experiments using flat microchambers. Simulations in cells with normal radius are shown in Movie S19. We have revised the movie descriptions to clarify this point.

I. Line numbers will help the reviewers.

We thank the reviewer for this suggestion and have included line numbers in the revised SI and main text.

Reviewer #1 (Remarks to the Author):

While the Min system has been extensively studied in the past, the study presented in this manuscript provides important new insights regarding the mechanisms that give rise to the observed pattern formations in vivo and in vitro. The authors have addressed all of my previous concerns and I recommend publication.

Reviewer #2 (Remarks to the Author):

I want to thank the authors for addressing my concerns. Rewriting the introduction has significantly improved the manuscript and the findings of their study are now presented in a clear manner. I recommend publication of the manuscript.

Reviewer #3 (Remarks to the Author):

The rebuttal and the revised manuscript are satisfactory. The authors have addressed most of my questions in detail and clarified the technical terms in the manuscript or in the letter. It is understandable that technical difficulties are realistic to hinder some expected experiments to fully match the theoretical model. I support its publication.